# Precision-based causal inference modulates audiovisual temporal recalibration

Luhe Li[1]*, Fangfang Hong[2], Stephanie Badde[3], Michael S Landy[1,4]

[1]Department of Psychology, New York University, New York, United States; [2]Department of Psychology, University of Pennsylvania, Philadelphia, United States; [3]Department of Psychology, Tufts University, Medford, United States; [4]Center for Neural Science, New York University, New York, United States

## eLife Assessment

In this **valuable** study, Li et al., set out to understand the mechanisms of audiovisual temporal recalibration - the brain's ability to adjust to the latency differences that emerge due to different (distance-dependent) transduction latencies of auditory and visual signals - through psychophysical measurements and modeling. The analysis and specification of a formal model for this process provide **convincing** evidence to support a role for causal inference in recalibration.

**\*For correspondence:**
luhe.li@nyu.edu

**Competing interest:** The authors declare that no competing interests exist.

**Abstract** Cross-modal temporal recalibration guarantees stable temporal perception across ever-changing environments. Yet, the mechanisms of cross-modal temporal recalibration remain unknown. Here, we conducted an experiment to measure how participants' temporal perception was affected by exposure to audiovisual stimuli with constant temporal delays that we varied across sessions. Consistent with previous findings, recalibration effects plateaued with increasing audiovisual asynchrony (nonlinearity) and varied by which modality led during the exposure phase (asymmetry). We compared six observer models that differed in how they update the audiovisual temporal bias during the exposure phase and in whether they assume a modality-specific or modality-independent precision of arrival latency. The causal-inference observer shifts the audiovisual temporal bias to compensate for perceived asynchrony, which is inferred by considering two causal scenarios: when the audiovisual stimuli have a common cause or separate causes. The asynchrony-contingent observer updates the bias to achieve simultaneity of auditory and visual measurements, modulating the update rate by the likelihood of the audiovisual stimuli originating from a simultaneous event. In the asynchrony-correction model, the observer first assesses whether the sensory measurement is asynchronous; if so, she adjusts the bias proportionally to the magnitude of the measured asynchrony. Each model was paired with either modality-specific or modality-independent precision of arrival latency. A Bayesian model comparison revealed that both the causal-inference process and modality-specific precision in arrival latency are required to capture the nonlinearity and asymmetry observed in audiovisual temporal recalibration. Our findings support the hypothesis that audiovisual temporal recalibration relies on the same causal-inference processes that govern cross-modal perception.

## Introduction

Perception is not rigid but rather can adapt to the environment. In a multimodal environment, misalignment across the senses can occur because signals in different modalities may arrive with different

physical and neural delays in the relevant brain areas (*Fain, 2019*; *Badde et al., 2020a*; *Pöppel, 1988*; *Spence and Squire, 2003*). Perceptual misalignment can also arise from changes in the environment, such as when wearing a virtual-reality headset or adapting to hearing aids. Cross-modal temporal recalibration serves as a critical mechanism to maintain perceptual synchrony despite changes in the perceptual systems and the environment (reviewed in *King, 2005*; *Vroomen and Keetels, 2010*). This phenomenon is exemplified in audiovisual temporal recalibration, where exposure to a consistent audiovisual stimulus-onset asynchrony (SOA) shifts the point of subjective simultaneity (PSS) between auditory and visual stimuli; as a result, stimuli at first perceived as temporally discrepant are gradually perceived as more synchronous (*Di Luca et al., 2009*; *Fujisaki et al., 2004*; *Hanson et al., 2008*; *Harrar and Harris, 2008*; *Heron et al., 2007*; *Keetels and Vroomen, 2007*; *Navarra et al., 2005*; *Roach et al., 2011*; *Tanaka et al., 2011*; *Vatakis et al., 2007*; *Vatakis et al., 2008*; *Vroomen et al., 2004*; *Vroomen and Keetels, 2010*).

However, the mechanisms of cross-modal temporal recalibration remain unknown. Current models of audiovisual temporal recalibration either did not specify the recalibration process (*Di Luca et al., 2009* ; *Navarra et al., 2009*; *Yarrow et al., 2015*), or did not capture all characteristics of recalibration effects (*Roach et al., 2011*; *Sato and Aihara, 2011*; *Yarrow et al., 2015*). Specifically, audiovisual temporal recalibration shows two distinct characteristics: the amount of recalibration is nonlinear and asymmetric as a function of the SOA participants are adapted to (adapter SOA). The amount of recalibration is not a linear function of the adapter SOA, but instead plateaus at an SOA of approximately 100–300 ms (*Fujisaki et al., 2004*; *Vroomen et al., 2004*). Recalibration can also be asymmetrical: its magnitude differs depending on whether the visual or auditory stimulus leads during the exposure phase (*Fujisaki et al., 2004*; *O'Donohue et al., 2022*; *Van der Burg et al., 2013*).

Here, we propose a causal-inference model to explain the mechanism of audiovisual temporal recalibration. Causal inference is the process by which the observer determines whether multisensory signals originate from a common source and adjust their perceptual estimates accordingly (*Sato et al., 2007*; *Shams and Beierholm, 2010*; *Wei and Körding, 2009*). Bayesian models based on causal inference have been proposed to explain multisensory integration effects (*Körding et al., 2007*; *Sato et al., 2007*), and these models have been empirically validated in studies of spatial audiovisual and visual-tactile integration (*Badde et al., 2020b*; *Beierholm et al., 2009*; *Rohe and Noppeney, 2015*; *Wozny et al., 2010*). In the temporal domain, some studies have successfully used causal inference to model the integration of cross-modal relative timing, accurately predicting simultaneity judgments in audiovisual speech (*Magnotti et al., 2013*) and more complex scenarios involving one auditory and two visual stimuli (*Sato, 2021*).

In the context of cross-modal recalibration, causal inference is expected to play a role based on the intuition that the amount of recalibration should be reduced when the multisensory signals are not perceived as causally related (*Fujisaki et al., 2004*; *Hsiao et al., 2022*; *Vroomen et al., 2004*). Supporting this, causal-inference models successfully predicted cross-modal spatial recalibration of visual-auditory (*Hong, 2023*; *Hong et al., 2021*; *Sato et al., 2007*) and visuo-tactile (*Badde et al., 2020b*) signals. Building on this framework, here, we propose a causal-inference model for cross-modal temporal recalibration that derives the multisensory percept based on inferences about the shared origin of the signals and updates the cross-modal temporal biases such that subsequent measurements are shifted toward the percept.

The first aim of this study is to test whether causal inference is necessary to explain the nonlinearity of audiovisual temporal recalibration across different adapter SOAs. To this aim, we compared the causal-inference model with two alternatives: an asynchrony-contingent model and an asynchrony-correction model. The asynchrony-contingent model scales the amount of recalibration by the likelihood that the sensory measurement of SOA was caused by a synchronous audiovisual stimulus pair. The model predicts a nonlinear recalibration effect across adapter SOAs without requiring observers to perform full Bayesian inference. The asynchrony-correction model assumes that recalibration only occurs when asynchronous onsets between the cross-modal stimuli are registered, followed by the update of the cross-modal temporal bias to compensate for this SOA measurement. This account is based on the intuitive rationale that repeated measurements of asynchrony can prompt the perceptual system to restore coherence. Unlike the causal-inference and the asynchrony-contingent models, this model predicts minimal recalibration when the adapter SOA falls within the range of measured

asynchronies that can arise with simultaneously presented stimuli due to sensory noise. This model serves as the baseline for model comparison.

The second aim was to examine factors that had the potential to drive the asymmetry of recalibration across visual-leading and auditory-leading adapter SOAs. It has been suggested that the asymmetry may be explained by physical and neural latency differences between signals (*O'Donohue et al., 2022*; *Van der Burg et al., 2013*). These latency differences can vary significantly based on the physical distance between the stimulus and the sensors, as well as the neural transmission time required for the signal to reach the relevant sensory region (*Hirsh and Sherrick, 1961*; *King, 2005*). Yet–speaking against a merely physical origin–these latency differences vary with sensory experience in infancy (*Badde et al., 2020a*). While these latency differences can explain the audiovisual temporal bias observed in most humans, they would affect recalibration to different adapter SOAs equally, making it unlikely for any asymmetry to arise. In contrast to latency differences, sensory uncertainty has been shown to affect the degree of cross-modal recalibration in a complex fashion (*Badde et al., 2020b*; *Hong et al., 2021*). We hypothesized that the difference across modalities in the precision of the arrival times, the time it takes visual and auditory signals to arrive in the relevant brain areas, plays a critical role in the asymmetry of cross-modal temporal recalibration.

To examine the mechanism underlying audiovisual temporal recalibration, we manipulated the adapter SOA across sessions, introducing asynchronies up to 0.7 seconds of either auditory or visual lead. Before and after the exposure phase in each session, we measured participants' perception of audiovisual relative timing using a ternary temporal-order judgment (TOJ) task. To preview the empirical results, we confirmed the nonlinearity of the recalibration effect: recalibration magnitude increased linearly for short adapter SOAs, but then reached an asymptote or even decreased with increasing adapter SOAs. Furthermore, participants showed idiosyncratic asymmetries of the recalibration effect across modalities; for most participants, the amount of recalibration was larger when the auditory stimulus led than when it lagged, but the opposite was found for other participants. To scrutinize the factors that might drive the nonlinearity and asymmetry of audiovisual temporal recalibration, we fitted six models to the TOJ data. These models determined the amount of recalibration based on either perceptual causal-inference processes, a heuristic evaluation of the common cause of the audiovisual stimuli, or a fixed criterion for the need to correct asynchrony. For each of these three models, we implemented either modality-specific or modality-independent precision of the arrival

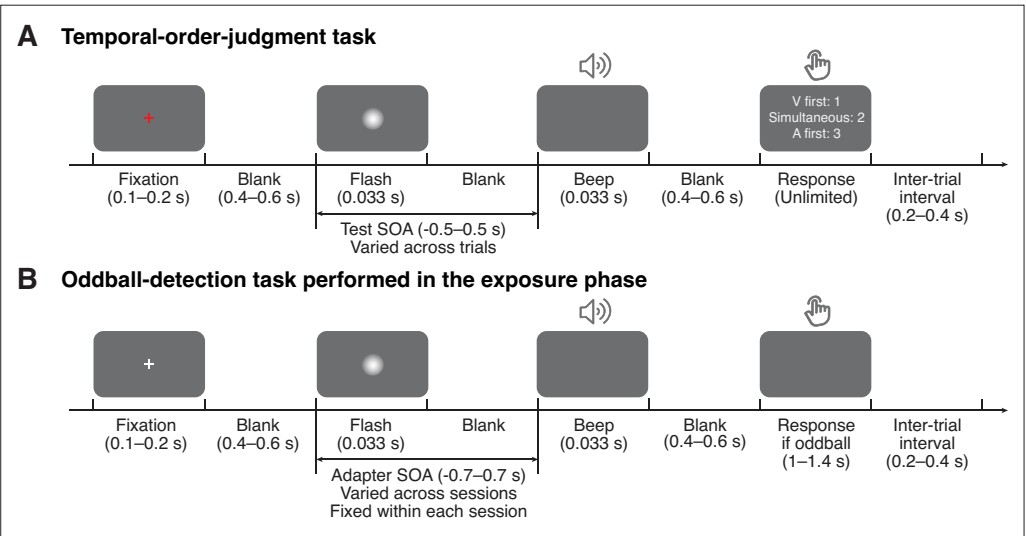

**Figure 1.** Task timing. (**A**) Temporal-order judgment task administered in the pre- and post-tests. In each trial, participants made a temporal-order judgment in response to an audiovisual stimulus pair with a varying stimulus-onset asynchrony (SOA). Negative values: auditory lead; positive values: visual lead. The contrast of the visual stimulus was enhanced for illustration purposes. (**B**) Oddball-detection task performed in the exposure phase and top-up trials during the post-test phase. Participants were repeatedly presented with an audiovisual stimulus pair with an SOA that was fixed within each session but varied across sessions. Occasionally, the intensity of either one or both of the stimuli was increased. Participants were instructed to, whenever an oddball stimulus appeared, press a key indicating the modality of the oddball (visual, auditory, or both) .

times. The model comparison revealed that the assumptions of Bayesian causal inference combined with modality-specific precision are essential to accurately capture the nonlinearity and idiosyncratic asymmetry of audiovisual temporal recalibration.

## Results

### Behavioral results

We adopted a classical three-phase recalibration paradigm in which participants completed a pre-test, an exposure phase, and a post-test in each session. In pre- and post-tests, we measured participants' perception of audiovisual relative timing using a ternary TOJ task: participants reported the perceived order ("visual first", "auditory first", or "simultaneous") of audiovisual stimulus pairs with varying SOA (range: from –0.5 to 0.5 s with 15 levels; *Figure 1A*). In the exposure phase, we induced temporal recalibration. Specifically, participants were exposed to a series of audiovisual stimuli with a consistent SOA (250 trials; *Figure 1B*). To ensure that participants were attentive to the stimuli, we inserted oddball stimuli with greater intensity in either one or both modalities (5% of the total trials independently sampled for each modality). Participants were instructed to press a key corresponding to the auditory, visual, or both oddballs whenever an oddball stimulus appeared. The high $d'$ of oddball-detection performance (auditory $d' = 3.34 \pm 0.54$, visual $d' = 2.44 \pm 0.72$) indicates that participants paid attention to both modalities. The post-test was almost identical to the pre-test, except that before every TOJ trial, there were three top-up oddball-detection trials to maintain the recalibration effect. In total, participants completed nine sessions on separate days. The adapter SOA (range: –0.7 to 0.7 s) was fixed within each session, but varied randomly across sessions and in random order across participants.

We compared the TOJs between the pre- and post-tests to examine the amount of audiovisual temporal recalibration induced by the audiovisual stimuli during the exposure phase. Specifically, we fitted the data from the pre- and post-tests jointly, allowing the curves to shift between the two tests while assuming the same shape for the psychometric functions that is determined by the relative arrival latencies, their precision, and fixed response criteria (*Figure 2A*; see Appendix 1 for the formalization

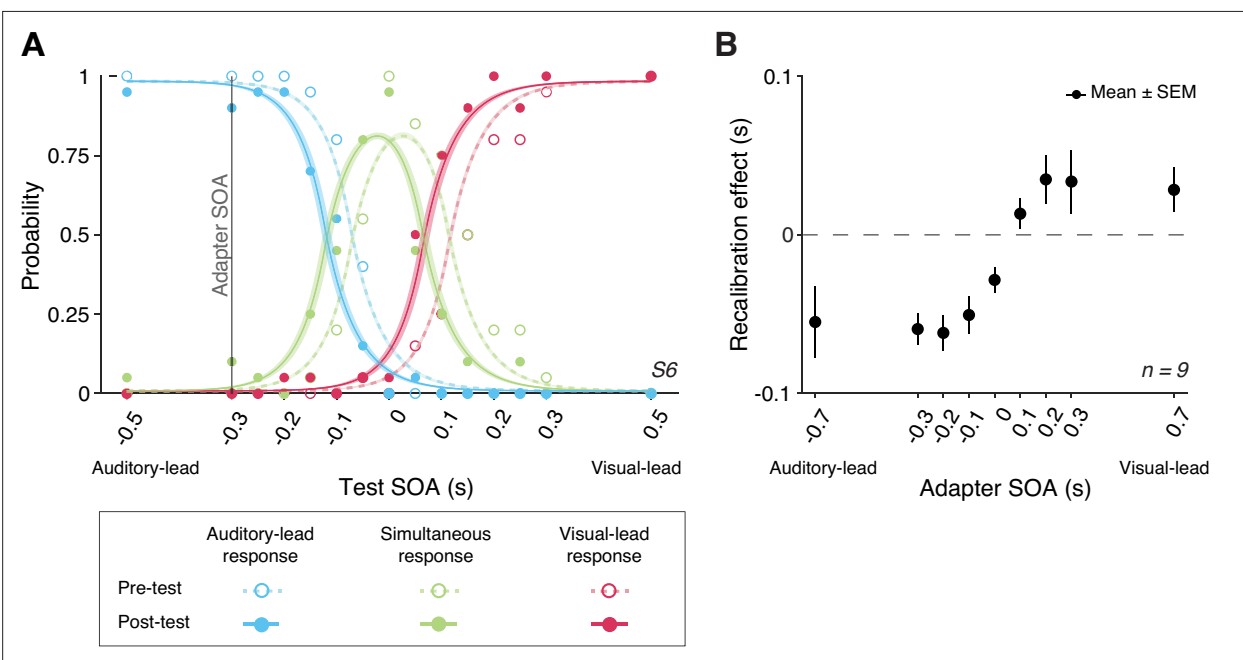

**Figure 2.** Behavioral results. (**A**) The probability of reporting that the auditory stimulus came first (blue), the two arrived at the same time (green), or the visual stimulus came first (red) as a function of test stimulus-onset asynchrony (SOA) for a representative participant in a single session. The adapter SOA was –0.3 s for this session. Curves: best-fitting psychometric functions estimated jointly using the data from the pre-test (dashed) and post-test (solid). Shaded areas: 95% bootstrapped confidence intervals. (**B**) Mean recalibration effects averaged across all participants as a function of adapter SOA. The recalibration effects are defined as the shifts in the point of subjective simultaneity (PSS) from the pre- to the post-test, where the PSS is the physical SOA at which the probability of reporting simultaneity is maximized. Error bars: ± SEM.

of the atheoretical model and an alternative model assuming a shift in the response criteria due to recalibration). The point of subjective simultaneity (PSS) is the physical SOA that corresponds to the maximum probability of reporting simultaneity (*Sternberg and Knoll, 1973*). The amount of audiovisual temporal recalibration was defined as the difference between the pre- and post-test PSSs. At the group level, we observed a nonlinear pattern of recalibration as a function of the adapter SOA: the amount of recalibration in the direction of the adapter SOA first increased but then plateaued with increasing magnitude of the adapter SOA (*Figure 2B*). Additionally, we observed an asymmetry in the amount of recalibration between auditory-leading and visual-leading adapter SOAs, with auditory-leading adapter SOAs inducing a greater amount of recalibration (*Figure 2B*; see *Appendix 2—figure 1A* for individual participants' data). To quantify this asymmetry for each participant, we calculated an asymmetry index, defined as the sum of the recalibration effects across all adapter SOAs (zero: no evidence for asymmetry; positive values: greater recalibration given visual-lead adapter SOAs; negative: greater recalibration given auditory-lead adapter SOAs). For each participant, we bootstrapped the TOJ responses to obtain a 95% confidence interval for the asymmetry index. Eight out of nine participants showed an asymmetry index significantly different from zero, with the majority showing greater recalibration for auditory-leading adapter SOAs, suggesting a general asymmetry in recalibration (*Appendix 2—figure 1B*).

## Modeling results

In the following sections, we describe our models for cross-modal temporal recalibration by first laying out the general assumptions of these models, and then elaborating on the differences between them. Then, we compare the models' ability to capture the observed data.

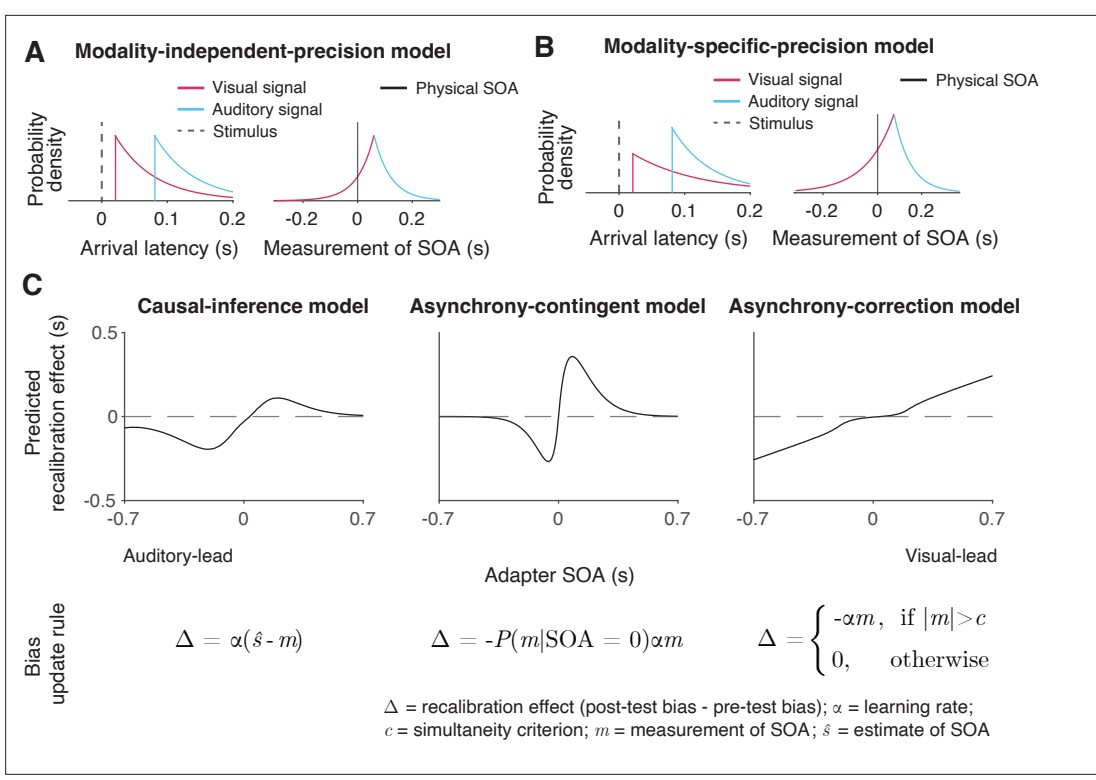

**Figure 3.** Illustration of the six observer models of cross-modal temporal recalibration. (**A**) Left: Arrival-latency distributions for auditory (cyan) and visual (pink) sensory signals. When the precision of arrival latency is modality-independent, these two exponential distributions have identical shape. Right: The resulting symmetrical double-exponential measurement distribution of the stimulus-onset asynchrony (SOA) of the stimuli. (**B**) When the precision of the arrival latencies is modality-specific, the arrival-latency distributions for auditory and visual signals have different shapes, and the resulting measurement distribution of the SOA is asymmetrical. (**C**) Bias update rules and predicted recalibration effects for the three contrasted recalibration models: The causal-inference model updates the audiovisual bias based on the difference between the estimated and measured SOA. The asynchrony-contingent model updates the audiovisual bias by a proportion of the measured SOA and modulates the update rate by the likelihood that the measured sensory signals originated from a simultaneous audiovisual pair. The asynchrony-correction model adjusts the audiovisual bias by a proportion of the measured SOA when this measurement exceeds fixed criteria for simultaneity.

## General model assumptions

We formulated six process models of cross-modal temporal recalibration (*Figure 3*). These models share several assumptions about audiovisual temporal perception and recalibration that we selected based on a comparison of atheoretical, descriptive models of TOJ data (Appendix 1). First, when an auditory and a visual signal are presented, the corresponding neural signals arrive in the relevant brain areas with a variable latency due to internal and external noise. We assume arrival times for the two modalities are independent and that the arrival latencies are exponentially distributed (*García-Pérez and Alcalá-Quintana, 2012*; *Figure 3A*, left panel). Moreover, we assume a constant offset between auditory and visual arrival times, reflecting an audiovisual temporal bias. A simple derivation shows that the resulting measurement of SOA has a double-exponential distribution (*Figure 3A*, right panel; see derivation in Appendix 3). The probability density function peaks at an SOA that is the physical SOA of the stimuli plus the observer's audiovisual temporal bias. The slopes of the measurement distribution reflect the precision of the arrival times; the steeper the slope, the more precise the measured latency. When the precision differs between modalities, the measurement distribution of the SOA between the auditory and visual stimuli is asymmetrical (*Figure 3B*).

Second, these models define temporal recalibration as the accumulation of updates to the audiovisual temporal bias after each encounter with an SOA. The accumulated shift in the audiovisual bias at the end of the exposure phase is then carried over to the post-test phase and persists throughout. Lastly, the bias is assumed to be reset to the same initial value in the pre-test across all nine sessions, reflecting the stability of the audiovisual temporal bias over time (*Grabot and van Wassenhove, 2017*).

## Models of cross-modal temporal recalibration

The six models we tested differed in the mechanisms governing the updates of the audiovisual bias during the exposure phase as well as the modality specificity of arrival time precision.

We formulated a temporal variant of the spatial Bayesian causal-inference model of recalibration (*Badde et al., 2020b*; *Hong, 2023*; *Hong et al., 2021*; *Sato et al., 2007*) to describe the recalibration of the relative timing between cross-modal stimuli (*Figure 3C*, left panel). In this model, when an observer is presented with an audiovisual stimulus pair during the exposure phase, they compute two intermediate estimates of the SOA between the stimuli, one for the common-cause scenario and the other for the separate-causes scenario. In the common-cause scenario, the estimated SOA of the stimuli is smaller than the measured SOA as it is combined with a prior distribution over SOA that reflects simultaneity. In the separate-causes scenario, the estimated SOA is approximately equal to the measured SOA. The two estimates are then averaged with each one weighted by the posterior probability of the corresponding causal scenario. The audiovisual bias is then updated to reduce the difference between the measured SOA and the combined estimate of the SOA. In other words, causal inference regulates the recalibration process by shifting the measured SOA to more closely match the percept, which in turn is computed based on the inferred causal structure.

The asynchrony-contingent model assumes that the observer estimates the likelihood that the sensory signals originated from a simultaneous audiovisual pair and updates the audiovisual bias to compensate for the measured SOA but the degree of compensation is scaled by this likelihood (*Figure 3C*, middle panel). There is a key distinction between the likelihood of simultaneity and the posterior probability of a common cause. The posterior probability of a common cause incorporates the prior distribution of SOAs when signals originate from the same source, including non-zero probabilities for SOAs ≠ 0. In contrast, the likelihood of simultaneity exclusively considers the case when SOA = 0. We assume that asynchrony-contingent observer computes the likelihood of simultaneity based on the knowledge of the double-exponential measurement distribution, instead of assuming a Gaussian measurement distribution as was done previously (*Maij et al., 2009*). The update rate of the audiovisual bias is proportional to this likelihood. For a stimulus pair with a large SOA, the average likelihood of the stimuli being physically simultaneous decreases, leading to reduced recalibration effects compared to stimulus pairs with smaller SOAs. Thus, this asynchrony-contingent model is capable of replicating the nonlinearity of recalibration across adapter SOAs without requiring the observer to perform full Bayesian inference.

The asynchrony-correction model assumes that the observer first compares the sensory measurement of SOA to their criteria for audiovisual simultaneity to decide whether to recalibrate in a given

trial. If the measured SOA falls within the range perceived as simultaneous according to the fixed criteria, the observer might attribute a non-zero measurement of SOA to sensory noise and omit recalibration. On the other hand, if the measured SOA exceeds this range, the observer perceives the stimuli as asynchronous, and shifts the audiovisual bias by a proportion of the measurement of SOA (*Figure 3C*, right panel). This model serves as a direct contrast to the causal-inference model, as it predicts an opposite pattern: a nonlinear but monotonic increase in the amount of temporal recalibration, with minimal recalibration when the measured SOA falls within the simultaneity range and increasing recalibration as the measured SOA moves further outside of this range.

We additionally assumed either modality-specific or modality-independent precision of the arrival times. Each choice suggests a different origin of the variability. The variability in arrival times is either modality specific, limited by neural-latency noise in each sensory channel (*Yarrow et al., 2022*), or modality independent, resulting from variabilities in a central timing mechanism (*Hirsh and Sherrick, 1961*).

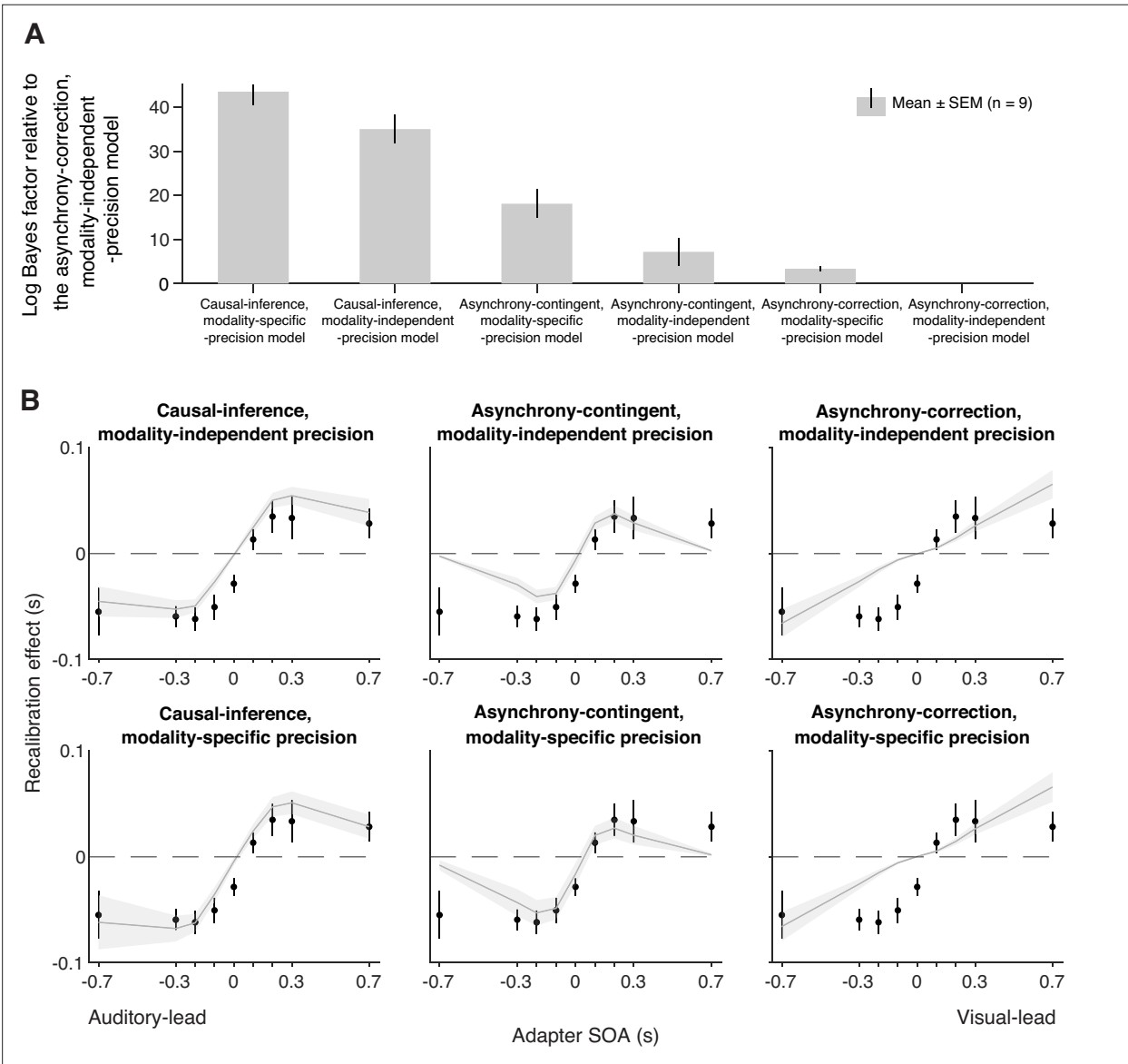

**Figure 4.** Model comparison and predictions. (**A**) Model comparison based on model evidence. Each bar represents the group-averaged log Bayes factor of each model relative to the asynchrony-correction, modality-independent-precision model, which had the weakest model evidence. (**B**) Empirical data (points) and model predictions (lines and shaded regions) for the recalibration effect as a function of adapter stimulus-onset asynchrony (SOA).

## Model fitting and model comparison

We fit six models to each participant's data. Each model was constrained jointly by the temporal-order judgments from the pre- and post-tests of all nine sessions. To quantify model performance, we calculated model evidence, i.e., the likelihood of each model given the data marginalized over all possible parameters, which revealed that the causal-inference model had the strongest model evidence at the group level and best fit the data of most participants, followed by the asynchrony-contingent model and then the asynchrony-correction model. To quantify the differences between model performance, we performed a Bayesian model comparison by computing the Bayes factor for each model relative to the worst-performing model, the asynchrony-correction model with modality-independent arrival-latency precision (*Figure 4A*, see *Appendix 4—figure 1* for individual-level model comparison). Within each of these three model categories, the version incorporating modality-specific precision consistently outperformed the modality-independent version.

## Model prediction

To inspect the quality of the model fit, for every model, we used the best-fitting parameter estimates for each participant to predict the group-average recalibration effect as a function of adapter SOA (*Figure 4B*). The nonlinearity of audiovisual temporal recalibration across adapter SOAs was captured by both the asynchrony-contingent and causal-inference models. Nonetheless, the causal-inference model outperformed the asynchrony-contingent model by accurately predicting a non-zero average recalibration effect at adapter SOAs of 0.7 and –0.7 s, where the asynchrony-contingent model predicted no recalibration. Additionally, incorporating modality-specific precision enabled both the asynchrony-contingent and causal-inference models to more accurately predict increased recalibration when the adapter SOA was auditory-leading. Overall, the model that relies on causal inference during the exposure phase and assumes modality-specific precision of arrival times most accurately captured both the nonlinearity and asymmetry of the recalibration effect. This model could also account for individual participants' idiosyncratic asymmetry in temporal recalibration to auditory- and visual-leading adapter SOAs (see *Appendix 5—figures 1–3* for predictions of individual participants' recalibration effects of all models; see *Appendix 6—figures 1–3* for predictions of individual participants' TOJ responses using the causal-inference models with modality-specific precision).

## Model simulation

Simulations with the causal-inference model revealed which factors of the modeled recalibration process determine the degree of nonlinearity and asymmetry of cross-modal temporal recalibration to different adapter SOAs. The prior belief that the auditory and visual stimuli share a common cause plays a crucial role in adjudicating the relative influence of the two causal scenarios (*Figure 5A*). When the observer has a prior belief that audiovisual stimuli always originate from the same source, they recalibrate by a proportion of the perceived SOA no matter how large the measured SOA is, mirroring the behavior of the asynchrony-correction model when its criteria for simultaneity are such that no stimuli are treated as simultaneous. On the other hand, when the observer believes that the audiovisual stimuli always have separate causes, they treat the audiovisual stimuli as independent of each other and do not recalibrate. Estimates of the common-cause prior for our participants fall between the two extreme beliefs, resulting in the nonlinear pattern of recalibration that lies between the extremes of no recalibration and the proportional recalibration effects as a function of the adapter SOA (see *Appendix 6—table 1* for parameter estimates for individual participants).

Simulations also identified key model elements of the causal-inference model that predict a non-zero recalibration effect even at large SOAs, a feature that distinguishes the causal-inference from the asynchrony-contingent model. This non-zero recalibration effect for large adapter SOAs can be replicated by either assuming a strong prior for a common cause (*Figure 5A*) or by assuming low sensory precision of arrival times (*Figure 5B*). Both relationships are intuitive: observers with a stronger prior belief in a common cause and ideal observers with lower sensory precision are more likely to assign a higher posterior probability to the common-cause scenario, leading to greater recalibration. A decrease of the spread of the prior distribution over SOA conditioned on a common cause increases the recalibration magnitude, but only over a small range of SOAs for which there is a higher probability of the common-cause scenario (*Appendix 7—figure 1A*), and thus cannot account for non-zero recalibration for large SOAs.

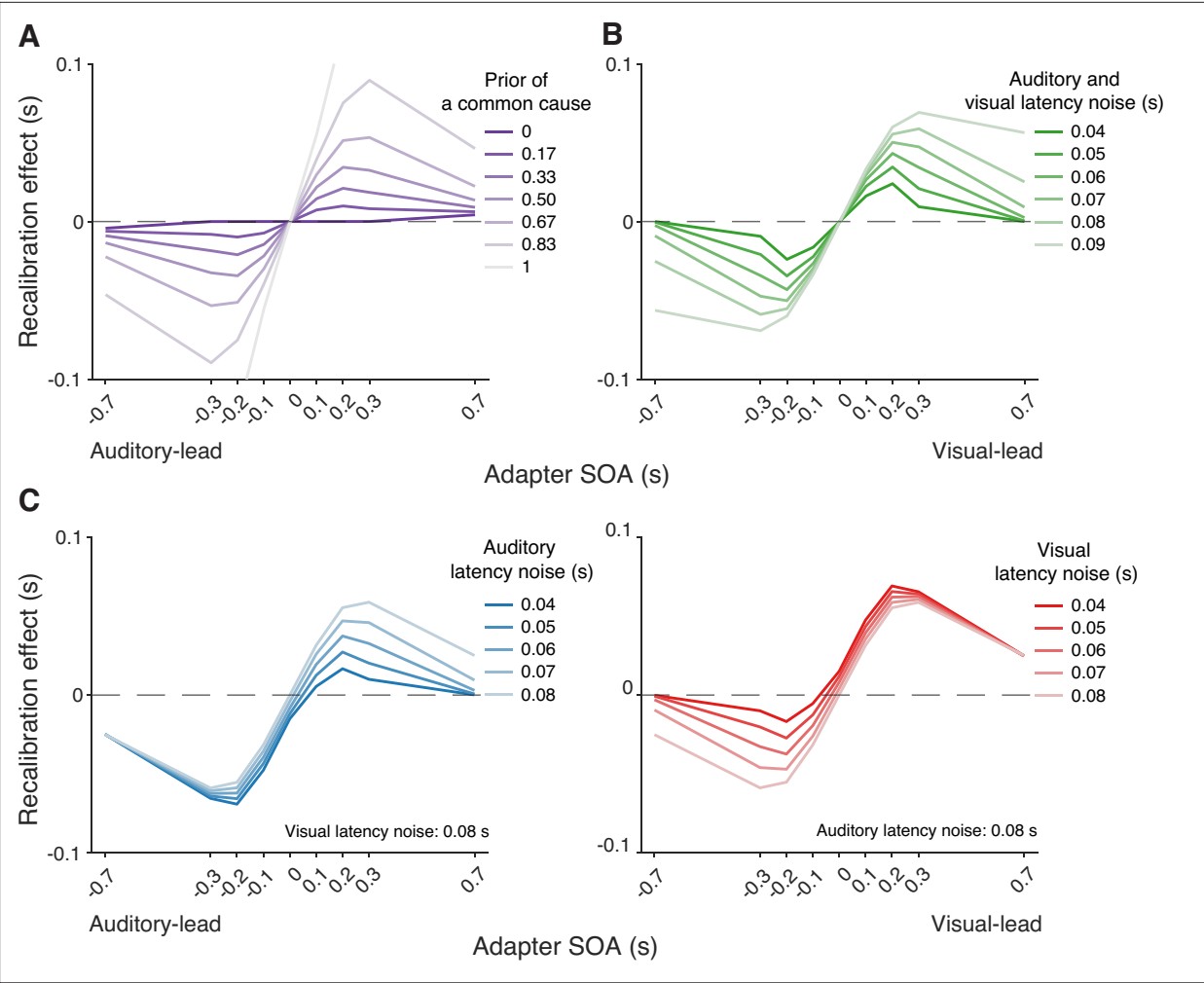

**Figure 5.** Simulation of temporal recalibration using the causal-inference model. (**A**) The influence of the observer's prior belief of a common cause: the stronger the prior, the larger the recalibration effects. (**B**) The influence of latency noise: recalibration effects increase with decreasing sensory precision (i.e. increasing latency noise captured by the exponential time constant) of both modalities. (**C**) The influence of auditory/visual latency noise: recalibration effects are asymmetric between auditory-leading and visual-leading adapter stimulus-onset asynchronies (SOAs) due to differences in the precision of auditory and visual arrival latencies. Left panel: Increasing auditory latency precision (i.e. reducing auditory latency noise) reduces recalibration in response to visual-leading adapter SOAs. Right panel: Increasing visual precision (i.e. reducing visual latency noise) reduces recalibration in response to auditory-leading adapter SOAs.

Differences in arrival-time precision between audition and vision result in an asymmetry of audio-visual temporal recalibration across adapter SOAs (*Figure 5C*). The amount of recalibration is attenuated when the modality with the higher precision lags the less precise one during the exposure phase. When the more recent stimulus component in a cross-modal pair is more precise, the observer is more likely to attribute the asynchrony to separate causes and thus recalibrate less. In addition, the fixed audiovisual bias does not affect asymmetry, but shifts the recalibration function laterally and determines the adapter SOA for which no recalibration occurs (*Appendix 7—figure 1B*).

## Discussion

This study scrutinized the mechanism underlying audiovisual temporal recalibration. We measured the effects of exposure to audiovisual stimulus pairs with a constant temporal offset (adapter SOA) on audiovisual temporal-order perception across a wide range of adapter SOAs. Recalibration effects changed nonlinearly with the magnitude of adapter SOAs and were asymmetric across auditory-leading and visual-leading adapter SOAs. We then compared the predictions of different observer

models for the amount of recalibration as a function of adapter SOA. A Bayesian causal-inference model with modality-specific precision of the arrival latencies fit the observed data best. These findings suggest that human observers rely on causal-inference-based percepts to recalibrate cross-modal temporal perception. These results align closely with studies that have demonstrated the role of causal inference in audiovisual (*Hong et al., 2021*) and visual-tactile spatial recalibration (*Badde et al., 2020b*). Our results are also consistent with previous recalibration models that assumed a strong relation between perception and recalibration (*Sato and Aihara, 2011*; *Sato et al., 2007*). Hence, we suggest that the same mechanisms underlie cross-modal perception and recalibration across different sensory features.

The observed recalibration results could not be predicted by the asynchrony-contingent model that employed a heuristic approximation of the causal-inference process. Even though this model was capable of predicting a nonlinear relationship between the recalibration effect and the adapter SOA, it failed to capture a non-zero recalibration effect at large adapter SOAs, as observed in several participants. This is becausethis model uses the likelihood of a synchronous audiovisual stimulus pair given the measured SOA to modulate the update rate of audiovisual bias, which will be very small on average for large SOAs. Therefore, the model predicts little to no recalibration at large adapter SOAs. In contrast, the causal-inference model can capture the non-zero recalibration effect because the common-cause scenario always influences the amount of recalibration even when the adapter SOA is too large to be perceived as synchronous. Simulation (*Figure 5A and B*) shows that a strong prior belief in a common cause or less precision of arrival times can result in non-zero recalibration effects following exposure to clearly asynchronous stimulus pairs. Notably, even though it might at first seem counter-intuitive that cross-modal temporal recalibration can be elicited by clearly asynchronous streams of sensory information, many of us have experienced this effect during laggy, long video conferences.

The asynchrony-correction model assumes that observers recalibrate to restore temporal synchrony whenever the SOA measurement indicates a temporal discrepancy, but this model predicts recalibration effects across adapter SOAs that are contrary to our observations. This suggests that cross-modal temporal recalibration is not merely triggered by an asynchronous sensory measurement of SOA and an attempt to correct it. In contrast, the causal-inference model accurately captured the plateau of the recalibration effects as adapter SOA increased, because the probability that the auditory and visual stimuli have separate causes also increased. This resulted in a smaller discrepancy between the sensory measurement and the final percept of the SOA, leading to less recalibration.

We found that most of our participants exhibited larger recalibration effects in response to exposure to audiovisual stimuli with a consistent auditory lead compared to exposure to a visual lead. This result is consistent with a previous study that reported greater cumulative recalibration in response to audiovisual stimuli with an auditory lead at the group level (*O'Donohue et al., 2022*). Our simulation results further suggested that this asymmetry in recalibration effects might be due to higher precision of auditory compared to visual arrival latencies. A few participants showed the opposite pattern: stronger recalibration effects following exposure to visual-leading audiovisual stimuli. This is not surprising, as causal-inference models often reveal substantial individual differences in sensory noise (*Hong et al., 2021*; *Magnotti et al., 2013*). A recent EEG study further provided neural correlates for individual sensory noise by identifying correlations between neural-latency noise and behavioral sensory noise measured from simultaneity-judgment tasks for audiovisual, visuo-tactile, and audio-tactile pairs (*Yarrow et al., 2022*). Therefore, our model explains how individual differences in precision of arrival latency could contribute to the asymmetry in cross-modal temporal recalibration observed in previous studies. For example, *Fujisaki et al., 2004*, found a slightly larger recalibration in response to audiovisual stimuli with a visual lead compared to an auditory lead, while their pilot results with the same design but a wider range of adapter SOAs showed the opposite pattern.

In order to incorporate causal inference into our recalibration models, we modeled recalibration as a shift of audiovisual bias. Building on previous latency-shift models (*Di Luca et al., 2009*; *Navarra et al., 2009*), we specified a mechanism for how the audiovisual bias is updated during the exposure to an audiovisual SOA. Our model is not mutually exclusive with other models that implement recalibration as a shift of simultaneity criteria (*Yarrow et al., 2011a*; *Yarrow et al., 2015*), or a change of sensitivity to discriminate SOA (*Roseboom et al., 2015*). A possible implementation of recalibration at the circuitry level is given by models assuming that audiovisual offsets are encoded by populations

of neurons tuned to different SOAs. In these models, recalibration is the consequence of selective gain reduction of neurons tuned to SOAs similar to the adapter SOA (*Cai et al., 2012*; *Roach et al., 2011*; *Yarrow et al., 2015*). Simulations show that this model can predict nonlinear recalibration effects as a function of adapter SOA depending on the number of neurons and the range of preferred SOAs (Appendix 8). However, to capture the asymmetric recalibration effects depending on which modality leads, one needs to incorporate inhomogeneous neuronal selectivity, i.e., unequal tuning curves, for auditory-leading and visual-leading SOAs.

Causal inference may effectively function as a credit-assignment mechanism to enhance perceptual accuracy during recalibration. In sensorimotor adaptation, humans correct motor errors that are more likely attributed to their own motor system rather than to the environment (*Berniker and Körding, 2008*; *Wei and Körding, 2009*). In visuomotor adaptation, substantial temporal recalibration occurs in response to exposure to movement-leading SOAs but less so to visual-leading SOAs (*Rohde and Ernst, 2012*; *Rohde et al., 2014*), because only movement-leading SOAs can be interpreted as causally linked sensory feedback from a preceding movement.

Causal-inference-based recalibration can further solve the conundrum that humans, despite the ability for cross-modal temporal recalibration, show persistent temporal biases (*Grabot and van Wassenhove, 2017*). These audiovisual and visual-tactile temporal biases appear to be shaped by early sensory experience (*Badde et al., 2020a*) and seem to be resistant to recalibration. The persistence of these biases contradicts recalibration models that correct for the measured cross-modal asynchrony. Instead, our causal-inference-based models of recalibration include an assumption that recalibration eliminates the discrepancy between measured and inferred asynchrony, both of which are influenced by cross-modal biases.

Previous studies have probed the role of causal inference for temporal recalibration and perception by experimentally varying task-irrelevant cues to a shared origin of the cross-modal stimuli, with mixed results. Earlier studies found no significant change in temporal recalibration when altering the sound presentation method (headphones vs. a speaker) or switching the presentation ear (*Fujisaki et al., 2004*), nor did recalibration effects vary with the spatial alignment of the audiovisual stimulus pair (*Keetels and Vroomen, 2007*). However, subsequent studies provide evidence that spatial grouping influences temporal recalibration, with the PSS shifting toward the temporal relationship suggested by spatially co-located stimuli (*Heron et al., 2012*; *Yarrow et al., 2011b*). Others found that spatial cues (*Heron et al., 2007*; *Yuan et al., 2012*) and featural content cues (e.g., male or female audiovisual speech and high-pitch sounds) (*Roseboom and Arnold, 2011*; *Roseboom et al., 2013*; *Yuan et al., 2012*) are both determinants of cross-modal temporal recalibration. Recent studies on audiovisual integration have extended causal-inference models to account for both the spatial position and temporal discrepancy of audiovisual signals (*Hong, 2023*; *McGovern et al., 2016*). Conversely, perceived conflicts in task-irrelevant features of visual-haptic stimuli do not influence the integration of task-relevant features, suggesting that causal inference is feature-specific rather than pertaining to whole objects (*Badde et al., 2023*). These studies indicate that task-relevant spatial and temporal information is taken into account for causal inference and might play a critical role in cross-modal recalibration.

## Methods
### Participants
Ten students from New York University (three males; age: 24.4 ± 1.77; all right-handed) participated in the experiment. They all reported normal or corrected-to-normal vision. All participants provided written informed consent before the experiment and received $15 per hour as monetary compensation. The study was conducted in accordance with the guidelines laid down in the Declaration of Helsinki and approved by the New York University Institutional Review Board (reference number: IRB-2016-595). One out of 10 participants was identified as an outlier and therefore excluded from further data analysis (Appendix 9).

### Apparatus and stimuli
Participants completed the experiments in a dark and semi-sound-attenuated room. They were seated 1 m from an acoustically transparent, white screen (1.36 × 1.02 m, 68 × 52° visual angle) and

placed their head on a chin rest. An LCD projector (Hitachi CP-X3010N, 1024 × 768 pixels, 60 Hz) was mounted above and behind participants to project visual stimuli on the screen. The visual and auditory stimulus durations were 33.33 ms. The visual stimulus was a high-contrast (36.1 cd/m$^2$) Gaussian blob (SD: 3.6°) on a gray background (10.2 cd/m$^2$) projected onto the screen. The auditory stimulus was a 500 Hz beep (50 dB SPL) without a temporal hamming window due to its short duration, which was played by a loudspeaker located behind the center of the screen. Some visual and auditory stimuli were of higher intensity, the parameters of these stimuli were determined individually (see Intensity-discrimination task). We adjusted the timing of audiovisual stimulus presentation and verified the timing using an oscilloscope (PICOSCOPE 2204A).

## Procedure

The experiment consisted of nine sessions, which took place on nine separate days. In each session, participants completed a pre-test, an exposure, and a post-test phase in sequence. The adapter SOA was fixed within each session, but varied across sessions (±700, ±300, ±200, ±100, 0 ms). The order of the adapter SOA was randomized across participants, with sessions separated by at least 1 day. The intensities of the oddball stimuli were determined prior to the experiment for each participant using an intensity-discrimination task to equate the difficulty of detecting oddball stimuli between participants and across modalities.

### Pre-test phase

Participants completed a ternary TOJ task during the pre-test phase. Each trial started with a fixation cross (0.1–0.2 s, uniform distribution; *Figure 1*), followed by a blank screen (0.4–0.6 s, uniform distribution). Then, an auditory and a visual stimulus (0.033 s) were presented with a variable SOA. There were a total of 15 possible test SOAs (±0.5 s and from –0.3 to 0.3 s in steps of 0.05 s), with positive values representing visual lead and negative values representing auditory lead. Following stimulus presentation there was another blank screen (0.4–0.6 s, uniform distribution), and then a response probe appeared on the screen. Participants indicated by button press whether the auditory stimulus occurred before or after the visual stimulus, or whether the two were simultaneous. There was no time limit for the response, and response feedback was not provided. The inter-trial interval was 0.2–0.4 s (uniform distribution). Each test SOA was presented 20 times in pseudo-randomized order, resulting in 300 trials in total, divided into five blocks. Participants usually took around 15 minutes to finish the pre-test phase.

### Exposure phase

Participants completed an oddball-detection task during the exposure phase. In each trial, participants were presented with an audiovisual stimulus pair with a fixed SOA (adapter SOA). In 10% of trials, the intensity of either the visual or the auditory component (or both) was greater than in the other trials. Participants were instructed to press the corresponding button as soon as possible to indicate whether there was an auditory oddball, a visual oddball, or both stimuli were oddballs. The task timing (*Figure 1B*) was almost identical to the ternary TOJ task, except that there was a response time limit of 1.4 s. Prior to the exposure phase, participants practiced the task for as long as needed to familiarize themselves with the task. During this practice, they were presented with bimodal stimuli with the same adapter SOA used in the exposure phase. There were a total of 250 trials, divided into five blocks. At the end of each block, we presented a performance summary with the hit rate and false-alarm rate of each modality. Participants usually completed the exposure phase in about 15 minutes.

### Post-test phase

Participants completed the ternary TOJ task as well as the oddball-detection task during the post-test phase. Specifically, each TOJ was preceded by three top-up (oddball-detection) trials. The adapter SOA in the top-up trials was the same as that in the exposure phase to prevent dissipation of temporal recalibration (*Machulla et al., 2012*). Both visual and auditory $d'$ remained consistent from the exposure to post-test phases, indicating similar performance in the top-up trials to performance during the exposure phase (*Appendix 10—figure 1*). To facilitate task switching, the ITI between the last top-up trial and the following TOJ trial was longer (with the additional time jittered around 1 s). Additionally, the fixation cross was displayed in red to signal the start of a TOJ trial. As in the pre-test phase, there

were 300 TOJ trials (15 test SOAs × 20 repetitions) with the addition of 900 top-up trials, grouped into six blocks. At the end of each block, we provided a summary of the oddball-detection performance. Participants usually completed the post-test phase in around one hour.

### Intensity-discrimination task

This task was conducted to estimate the just-noticeable-difference (JND) in intensity for a standard visual stimulus with a luminance of 36.1 cd/m² and a standard auditory stimulus with a volume of 40 dB SPL. The task was two-interval, forced choice. The trial started with a fixation (0.1–0.2 s) and a blank screen (0.4–0.6 s). Participants were presented with a standard stimulus (0.033 s) in one randomly selected interval and a comparison stimulus (0.033 s) in the other interval, temporally separated by an inter-stimulus interval (0.6–0.8 s). They indicated which interval contained the brighter/louder stimulus without time constraint. Seven test stimulus levels (luminance range: 5–195% relative to the standard visual stimulus intensity; volume range: 50–150% relative to the standard auditory stimulus' amplitude) were repeated 20 times, resulting in 140 trials for each task. We fit a cumulative Gaussian distribution function to these data and defined the oddball as an auditory or visual stimulus with an intensity judged as more intense than the standard 90% of the time. A higher probability than the standard JND of 75% was selected because the pilot studies showed that the harder oddball-detection task became too demanding during the one-hour post-test.

## Modeling

In this section, we first outline general assumptions, shared across all candidate models, regarding sensory noise, measurements, and bias. Then, we formalize three process models of recalibration that differ in the implementation of recalibration. In each recalibration model, we also provide a formalization of the ternary TOJ task administered in the pre- and the post-test phases, data from which were used to constrain the model parameters. Finally, we describe how the models were fit to the data.

### General modal assumptions regarding sensory noise, measurements, and bias

When an audiovisual stimulus pair with an SOA, $s = t_A - t_V$, is presented, it triggers auditory and visual signals that are registered in the relevant region of cortex where audiovisual temporal-order comparisons are made. This leads to two internal measurements of the arrival time for each signal in

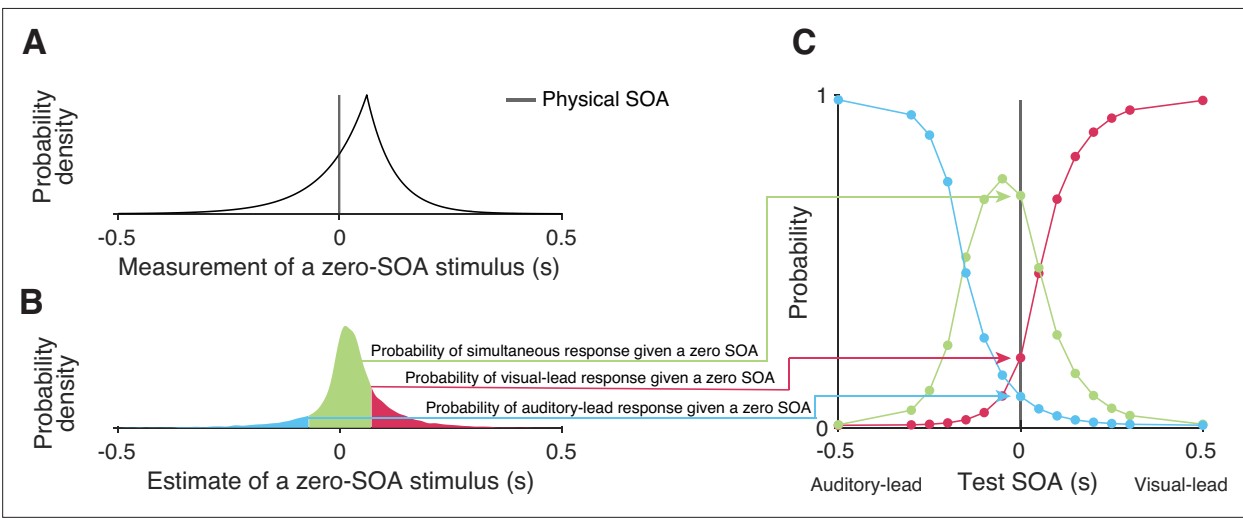

**Figure 6.** Simulating responses of the temporal-order-judgment (TOJ) task with a causal-inference perceptual process. (**A**) An example probability density for the measurement of a zero stimulus-onset asynchrony (SOA). (**B**) The probability density of estimates resulting from a zero-SOA stimulus based on simulation using the causal-inference process. The symmetrical criteria around zero partition the distribution of estimated SOA into three regions, coded by different colors. The area under each segment of the estimate distribution corresponds to the probabilities of the three possible intended responses for a zero SOA. (**C**) The simulated psychometric function computed by repeatedly calculating the probabilities of the three response types across all test SOAs.

an observer's brain. These arrival times are subject to noise and thus vary across presentations of the same physical stimulus pair. As in previous work (*García-Pérez and Alcalá-Quintana, 2012*), we model the probability distribution of the arrival time as shifted exponential distributions (*Figure 3A*). The arrival time of the auditory signal relative to onset $t_A$ is the sum of the fixed delay of internal signal, $\beta_A$, and an additional random delay that is exponentially distributed with time constant $\tau_A$; analogous for the visual latency (with delay $\beta_V$ and time constant $\tau_V$).

The measured SOA of the audiovisual stimulus pair is modeled as the difference of the arrival times of the two stimuli. Thus, the sensory measurement of SOA, $m$, reflects the sum of three components: the physical SOA, $s$; a fixed latency that is the difference between the auditory and visual fixed delay, $\beta_{pre} = \beta_A - \beta_V$; and the difference between two exponentially distributed random delays. A negative value of $\beta_{pre}$ indicates faster auditory processing. We assume that the audiovisual fixed latency corresponds to the observer's persistent default audiovisual temporal bias (*Badde et al., 2020a*; *Grabot and van Wassenhove, 2017*). Thus, we assume that after leaving the experimental room, the default bias is restored and thus consistent across pre-tests.

We model the recalibration process as a shift of the audiovisual temporal bias at the end of every exposure trial $i$, $\beta_i = \beta_{pre} + \Delta_{\beta,i}$, where $\beta_i$ is the current audiovisual bias, and $\Delta_{\beta,i}$ is the cumulative shift of audiovisual temporal bias. After the 250 exposure trials the updated biases can be expressed as $\beta_{post} = \beta_{pre} + \Delta_{\beta,250}$. We also assume that the amounts of auditory and visual latency noise, $\tau_A$ and $\tau_V$, remain constant across phases and sessions.

Given that both latency distributions are shifted exponential distributions, the probability density function of the sensory measurements of SOA, $m$, given physical SOA, $s$, is a double-exponential function (see derivation in Appendix 3; *Figure 6A*):

$$p(m_i | s_i, \beta_i) = \begin{cases} \dfrac{1}{\tau_A + \tau_V} \exp\left[\tau_V^{-1}(m_i - (s_i + \beta_i))\right], & \text{if } m_i \leq s_i + \beta_i, \\ \dfrac{1}{\tau_A + \tau_V} \exp\left[-\tau_A^{-1}(m_i - (s_i + \beta_i))\right], & \text{if } m_i > s_i + \beta_i. \end{cases} \quad (1)$$

The probability density function of measured SOA peaks at the physical SOA of the stimuli plus the participant's audiovisual temporal bias, $s_i + \beta_i$. The left and right spread of this measurement distribution depends on the amount of the latency noise for the visual, $\tau_V$, and auditory, $\tau_A$, signals. In models with modality-independent arrival-time precision, $\tau_A = \tau_V$ and the measurement distribution is symmetrical. This symmetrical measurement distribution is often approximated by a Gaussian distribution to fit TOJ responses in previous temporal-recalibration studies (*Di Luca et al., 2009*; *Fujisaki et al., 2004*; *Harrar and Harris, 2005*; *Keetels and Vroomen, 2007*; *Navarra et al., 2005*; *Tanaka et al., 2011*; *Vatakis et al., 2007*; *Vatakis et al., 2008*; *Vroomen et al., 2004*). Note that we assume the observer has perfect knowledge of the visual and auditory latency noise. Thus, the density of the measurement distribution corresponds to the likelihood function during the inference process when the observer only has the noisy measurement, $m$, and needs to infer the physical SOA, $s$.

## The causal-inference model
### Formalization of recalibration in the exposure phase
The causal-inference model assumes that, at the end of every exposure trial $i$, a discrepancy between the measured SOA, $m_i$, and the final estimate of the stimulus SOA, $\hat{s}_i$, signals the need for recalibration. The cumulative shift of audiovisual temporal bias $\Delta_{\beta,i}$ after exposure trial $i$ is

$$\Delta_{\beta,i+1} = \Delta_{\beta,i} + \alpha(\hat{s}_i - m_i), \quad (2)$$

where $\alpha$ is the learning rate.

The ideal observer infers intermediate location estimates for two causal scenarios: the auditory and visual stimuli can arise from a single cause ($C = 1$) or two independent causes ($C = 2$). The posterior distribution of the SOA, $s$, conditioned on each causal scenario is computed by multiplying the likelihood function (*Equation 1*) with the corresponding prior over SOA. In the case of a common cause ($C = 1$), the prior distribution of the SOA between sound and light is a Gaussian distribution (*Magnotti et al., 2013*; *McGovern et al., 2016*) $p(s|C = 1) = \mathcal{N}(0, \sigma_{C=1}^2)$. To maintain consistency with previous studies, we used an unbiased prior which assigns the highest probability to a physically synchronous

stimulus pair $s = 0$. Similarly, the prior distribution conditioned on separate causes ($C = 2$) is also a Gaussian distribution, $p(s|C = 2) = \mathcal{N}(0, \sigma^2_{C=2})$, with a much larger spread compared to the common-cause scenario. The intermediate estimates $\hat{s}_{C=1}$ conditioned on the common-cause scenario and $\hat{s}_{C=2}$ conditioned on separate-cause scenario are the maximum-a-posteriori estimates of conditional posteriors, which are approximated numerically as there is no closed-form solution.

The final estimate of the stimulus SOA, $\hat{s}$, depends on the posterior probability of each causal scenario. According to Bayes rule, the posterior probability that an audiovisual stimulus pair with the measured SOA, $m$, shares a common cause is

$$P(C = 1|m) = \frac{P(m|C = 1)P(C = 1)}{P(m|C = 1)P(C = 1) + P(m|C = 2)(1 - P(C = 1))}. \quad (3)$$

The likelihood of a common source/separate sources for a fixed SOA measurement was approximated by numerically integrating the scenario-specific protoposterior (i.e. the un-normalized posterior),

$$P(m|C = 1) = \int P(m|s)P(s|C = 1)ds,$$

$$P(m|C = 2) = \int P(m|s)P(s|C = 2)ds. \quad (4)$$

The posterior probability of a common cause additionally depends on the observer's prior belief of a common cause for auditory and visual stimuli, $P(C = 1) = p_{\text{common}}$.

The final estimate of SOA was derived by model averaging, i.e., the average of the scenario-specific SOA estimates, $\hat{s}_{C=1}$ and $\hat{s}_{C=2}$ each weighted by the posterior probability of the corresponding causal scenario,

$$\hat{s} = \hat{s}_{C=1}P(C = 1|m) + \hat{s}_{C=2}(1 - P(C = 1|m)). \quad (5)$$

## Formalization of the ternary TOJ task with a causal-inference perceptual process

In the ternary TOJ task administered in the pre- and post-test phases, the observer is presented with an audiovisual stimulus pair and has to decide whether the auditory stimulus was presented first, the visual stimulus was presented first, or both of them were presented at the same time. The observer makes this perceptual judgment by comparing the final estimate of the SOA, $\hat{s}$, to two internal criteria (*Cary et al., 2024*; *García-Pérez and Alcalá-Quintana, 2012*). We assume that the observer has a symmetric pair of criteria, $\pm c$, centered on the stimulus SOA corresponding to perceptual simultaneity ($\hat{s} = 0$). In addition, the observer may lapse or make an error when responding by a lapse rate, $\lambda$. The probabilities of reporting visual lead, $\Psi_V$, auditory lead, $\Psi_A$, or that the two stimuli were simultaneous, $\Psi_S$, are thus

$$\Psi_V(s) = \frac{\lambda}{3} + (1 - \lambda)\tilde{p}(\hat{s} > c|s),$$

$$\Psi_A(s) = \frac{\lambda}{3} + (1 - \lambda)\tilde{p}(\hat{s} < -c|s) \text{ and} \quad (6)$$

$$\Psi_S(s) = 1 - \Psi_V(s) - \Psi_A(s).$$

The probability distribution of causal-inference-based SOA estimates $p(\hat{s}|s)$ has no closed form distribution function and thus was approximated using simulations, resulting in $\tilde{p}(\hat{s}|s)$. *Figure 6* illustrates the process of simulating the psychometric functions, using a zero test SOA as an example. First, we sampled 10,000 SOA measurements from the double-exponential probability distribution corresponding to the test SOA of zero (*Figure 6A*). Second, for each sampled measurement, we simulated the process by which the observer carries out causal inference and by doing so produces an estimate of the stimulus SOA, while keeping the causal-inference model parameters fixed. This process resulted in a Monte Carlo approximation of the probability density distribution of the causal-inference-based SOA estimates (*Figure 6B*). Third, we calculated the probability of the three types

of responses (*Equation 6*) for this specific test SOA. This process was repeated for each test SOA to generate three psychometric functions (*Figure 6C*).

## The asynchrony-contingent model

In the asynchrony-contingent model, the observer measures the audiovisual SOA, $s$, by comparing the arrival latency of the auditory and visual signals. The observer uses the likelihood that the audiovisual stimuli occurred simultaneously $P(m|\text{SOA} = 0)$ to update the temporal bias during recalibration, instead of performing causal inference. We again assume that the observer has perfect knowledge about the precision and fixed delays of the arrival times and thus assume the likelihood corresponds to the measurement distribution (*Equation 1*). The observer uses this probability of simultaneity to scale the update rate of the audiovisual bias,

$$\Delta_{\beta,i+1} = \Delta_{\beta,i} - P(m_i|\text{SOA} = 0)\alpha m_i. \tag{7}$$

We assume the observer's estimate of the stimulus SOA, $\hat{s}$, is identical to the measured SOA, $m$. Thus, from the experimenter's perspective, the probability of the three different responses in the TOJ task can be obtained by replacing the SOA estimate, $\hat{s}$, with the SOA measurement, $m$, in *Equation 6*. As we know the probability distribution of $m$, the psychometric functions have a closed form (*García-Pérez and Alcalá-Quintana, 2012*).

## The asynchrony-correction model

In the asynchrony-correction model, the observer begins by evaluating if the sensory measurement of SOA, $m$, falls outside the criterion range for reporting that the two stimuli were presented simultaneously ±$c$. If the measurement does exceed this criterion, the observer adjusts the audiovisual bias by shifting it against the measurement, i.e., shifting it so that the measured SOA of a pair would be closer to zero and is more likely to perceived as simultaneous. This adjustment is proportional to the sensory measurement of the SOA, $m$, at a fixed rate determined by the learning rate $\alpha$. The update rule of the audiovisual bias in trial $i$ is thus

$$\Delta_{\beta,i+1} = \begin{cases} \Delta_{\beta,i} - \alpha m_i, & \text{if } |m_i| > c \\ \Delta_{\beta,i}, & \text{otherwise.} \end{cases} \tag{8}$$

The derivation of the psychometric functions is identical to the asynchrony-contingent model.

---

**Table 1.** Model parameters.

Check marks signify that the parameter is used for determining the likelihood of the data from the temporal-order judgment task in the pre- and post-test phase and/or for the Monte Carlo simulation of recalibration in the exposure phase.

| Notation | Specification | Temporal-order-judgment task | Recalibration in the exposure phase |
|---|---|---|---|
| $\beta_{\text{pre}}$ | The fixed relative delay between visual and auditory processing, i.e., the audiovisual bias prior to the exposure phase | ✓ | ✓ |
| $\tau_A$ | Amount of auditory latency noise, the exponential time constant of the auditory arrival-latency distribution | ✓ | ✓ |
| $\tau_V$ | Amount of visual latency noise, the exponential time constant of the visual-latency distribution | ✓ | ✓ |
| $\sigma_{C=1}$ | The spread of the Gaussian prior for the common-cause scenario | ✓ | ✓ |
| $\sigma_{C=2}$ | The spread of the Gaussian prior for the separate-causes scenario | ✓ | ✓ |
| $p_{\text{common}}$ | The prior probability of a common cause | ✓ | ✓ |
| $c$ | Simultaneity criterion | ✓ | |
| $\lambda$ | Lapse rate | ✓ | |
| $\alpha$ | Learning rate for shifting audiovisual bias | | ✓ |

---

## Model fitting

### Model log-likelihood

The model was fitted by optimizing the lower bound on the marginal log-likelihood. We fit the model to the joint ternary TOJ data collected during the pre- and post-test phases of all sessions. We did not collect TOJs in the exposure phase. But, to model the post-test data, we need to estimate the distribution of shifts of audiovisual bias resulting from the exposure phase ($\Delta_{\beta,250}$). We do this using Monte Carlo simulation of the 250 exposure trials to estimate the probability distribution of the cumulative shifts.

The set of model parameters $\Theta$ is listed in **Table 1**. There are $J$ sessions, each including $K$ trials in the pre-test phase and $K$ trials in the post-test phase. We denote the full dataset of pre-test data as $X_{\text{pre}}$ and for the post-test data as $X_{\text{post}}$. We fit the pre- and post-test data jointly by summing their log-likelihood, $\log p(X|M, \Theta) = \log p(X_{\text{pre}}|M, \Theta) + \log p(X_{\text{post}}|M, \Theta)$.

In a given trial, the observer responded either auditory-first (A), visual-first (V), or simultaneous (S). We denote a single response using indicator variables that are equal to 1 if that was the response in that trial and 0 otherwise. These variables for trial $k$ in session $j$ are $r^A_{\text{pre},jk}$, $r^V_{\text{pre},jk}$, and $r^S_{\text{pre},jk}$ for the pre-test trials, and $r^A_{\text{post},jk}$, etc., for the post-test trials. The log-likelihood of all pre-test responses $X_{\text{pre}}$ given the model parameters is

$$
\log p(X_{\text{pre}}|M, \Theta) = \sum_{j=1}^{J} \sum_{k=1}^{K} \left( r^A_{\text{pre},jk} \log \Psi_{A,\text{pre}}(s_{jk}) + \right.
$$

$$
\left. r^V_{\text{pre},jk} \log \Psi_{V,\text{pre}}(s_{jk}) + r^S_{\text{pre},jk} \log \Psi_{S,\text{pre}}(s_{jk}) \right). \tag{9}
$$

The psychometric functions for the pre-test (e.g., $\Psi_{A,\text{pre}}$) are defined in **Equation 6**, and are the same across all sessions as we assumed that the audiovisual bias $\beta_{\text{pre}}$ was the same before recalibration in every session.

The log-likelihood of responses in the post-test depends on the audiovisual bias after recalibration $\beta_{\text{post},j} = \beta_{\text{pre}} + \Delta_{\beta,250,j}$ for session $j$. To determine the log-likelihood of the post-test data requires us to integrate out the unknown value of the cumulative shift $\Delta_{\beta,250,j}$. We approximated this integral in two steps based on our previous work (**Hong et al., 2021**). First, we simulated the 250 exposure trials 1000 times for a given set of parameters $\Theta$ and session $j$. This resulted in 1000 values of $\Delta_{\beta,250,j}$. The distribution of these values was well fit by a Gaussian whose parameters were determined by the empirical mean and standard deviation of the sample distribution, resulting in the distribution $\tilde{p}(\Delta_{\beta,250,j}|M, \Theta)$. Second, we approximated the integral of the log-likelihood of the data over possible values of $\Delta_{\beta,250,j}$ by numerical integration. We discretized the approximated distribution $\tilde{p}(\Delta_{\beta,250,j}|M, \Theta)$ into 100 equally spaced bins centered on values $\Delta_{\beta,250,j}(n)$ ($n = 1, \cdots, 100$). The range of the bins was triple the range of the values from the Monte Carlo sample, so that the lower bound was $lb_{\Delta_{\beta,250,j}} = \Delta_{\beta,250,j,\min} - (\Delta_{\beta,250,j,\max} - \Delta_{\beta,250,j,\min})$ and the upper bound was $ub_{\Delta_{\beta,250,j}} = \Delta_{\beta,250,j,\max} + (\Delta_{\beta,250,j,\max} - \Delta_{\beta,250,j,\min})$.

The log-likelihood of the post-test data was approximated as

$$\log p(X_{\text{post}}|M,\Theta) = \sum_{j=1}^{J} \log \left( \int p(X_{\text{post}}|\Delta_{\beta,250,j},M,\Theta)p(\Delta_{\beta,250,j}|M,\Theta)d\Delta_{\beta,250,j} \right)$$

$$\approx \sum_{j=1}^{J} \log \left( \int_{lb_{\Delta_{\beta,250,j}}}^{ub_{\Delta_{\beta,250,j}}} p(X_{\text{post}}|\Delta_{\beta,250,j},M,\Theta) \right.$$

$$\left. \tilde{p}(\Delta_{\beta,250,j}|M,\Theta)d\Delta_{\beta,250,j} \right) \tag{10}$$

$$\approx \sum_{j=1}^{J} \log \left( \frac{ub_{\Delta_{\beta,250,j}} - lb_{\Delta_{\beta,250,j}}}{100} \sum_{n=1}^{100} p(X_{\text{post}}|\Delta_{\beta,250,j}(n),M,\Theta)\times \right.$$

$$\left. \tilde{p}(\Delta_{\beta,250,j}(n)|M,\Theta) \right),$$

where

$$p(X_{\text{post}}|\Delta_{\beta,250,j}(n),M,\Theta) = \prod_{k=1}^{K} \left( \Psi_{A,\text{post},jn}(s_{jk})^{r^A_{\text{post},jk}} \times \right.$$

$$\left. \Psi_{V,\text{post},jn}(s_{jk})^{r^V_{\text{post},jk}} \times \Psi_{S,\text{post},jn}(s_{jk})^{r^S_{\text{post},jk}} \right). \tag{11}$$

The psychometric functions in the post-test (e.g., $\Psi_{A,\text{post},jn}$) differed across sessions and bins because the simulated audiovisual bias after the exposure phase $\beta_{\text{post},j}$ depends on the adapter SOA fixed in session $j$ and the simulation bin $n$.

## Parameter estimation and model comparison

We approximated the lower bounds to the model evidence (i.e. the marginal likelihood) of each model for each participant's data using variational Bayesian Monte Carlo (*Acerbi, 2018*; *Acerbi, 2020*). We set the prior distribution of parameters based on the results of maximum-likelihood estimation using Bayesian Adaptive Direct Search to ensure that the parameter ranges were plausible (*Acerbi and Ma, 2017*). We repeated each search 20 times with a different and random starting point to address the possibility of reporting a local minimum. For each model, the fit with the maximum lower bounds of the model evidence across the repeated searches was chosen for the maximum model evidence and best parameter estimates.

We then conducted a Bayesian model comparison based on model evidence. The model with the strongest evidence was considered the best-fitting model (*MacKay, 2003*). To quantify the support of model selection, we computed the Bayes factor, the ratio of the model evidence between each model and the asynchrony-correction, modality-independent-precision model, which had the weakest model evidence. To compare any two models, one can simply calculate the difference in their log Bayes factors as both are relative to the same weakest model.

## Model recovery and parameter recovery

We conducted a model-recovery analysis for the six models and confirmed that they are identifiable (Appendix 11). In addition, we considered an alternative causal-inference model in which the bias update is proportional to the posterior probability of a common cause, instead of driven by the difference between the measured and estimated SOA. A separate model recovery analysis on variations of the causal-inference model was unable to distinguish between them (Appendix 12). For the causal-inference, modality-specific-precision model, we also carried out a parameter recovery analysis and confirmed that all the parameters are recoverable (Appendix 13).

## Acknowledgements

We thank the NYU High-Performance Computing (NYU HPC) for providing computational resources and support. We thank the anonymous reviewers' advice that helped us improve the manuscript. Funding: This research was funded by NIH EY08266.

# Additional information

### Funding

| Funder | Grant reference number | Author |
| --- | --- | --- |
| National Institutes of Health | EY08266 | Michael S Landy |

The funders had no role in study design, data collection and interpretation, or the decision to submit the work for publication.

### Author contributions

Luhe Li, Conceptualization, Data curation, Software, Formal analysis, Validation, Investigation, Visualization, Methodology, Writing – original draft; Fangfang Hong, Conceptualization, Resources, Supervision, Methodology, Writing – review and editing; Stephanie Badde, Conceptualization, Supervision, Methodology, Writing – review and editing; Michael S Landy, Conceptualization, Supervision, Funding acquisition, Methodology, Project administration, Writing – review and editing

### Author ORCIDs

Luhe Li ⓘ https://orcid.org/0000-0002-0182-3952
Fangfang Hong ⓘ https://orcid.org/0000-0003-1890-1977
Stephanie Badde ⓘ https://orcid.org/0000-0002-4005-5503
Michael S Landy ⓘ http://orcid.org/0000-0002-2079-4552

### Ethics

All participants provided informed written consent before the experiment and received monetary compensation. The study was conducted in accordance with the guidelines laid down in the Declaration of Helsinki and approved by the New York University Institutional Review Board.

Reviewer #1 (Public review): https://doi.org/10.7554/eLife.97765.3.sa1
Reviewer #2 (Public review): https://doi.org/10.7554/eLife.97765.3.sa2
Reviewer #3 (Public review): https://doi.org/10.7554/eLife.97765.3.sa3
Author response https://doi.org/10.7554/eLife.97765.3.sa4

# Additional files

### Supplementary files

MDAR checklist

### Data availability

All data and code are available via the Open Science Framework.

The following dataset was generated:

| Author(s) | Year | Dataset title | Dataset URL | Database and Identifier |
| --- | --- | --- | --- | --- |
| Li L, Hong F, Badde S, Landy MS | 2024 | Precision-based causal inference modulates audiovisual temporal recalibration | https://osf.io/8s7qv/ | Open Science Framework, 10.17605/OSF.IO/8S7QV |

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

## Appendix 1

### Comparing atheoretical models of audiovisual temporal recalibration

*Figure 2* in the main text shows raw psychometric data with fit psychometric functions (panel A) and estimates of PSS shift across sessions (panel B). These estimates are from fitting an *atheoretical* model that estimates PSS shift without reference to an underlying model of how recalibration comes about. Rather, it aims to show the pattern of recalibration from the data alone. Here, we describe how those estimates were computed and compare four different models of recalibration for this purpose. These four models vary based on two factors: the use of either a Gaussian or an exponential measurement distribution, and the assumption of either a bias shift or a criterion shift. Next, we describe these models followed by a model comparison.

### Gaussian vs. exponential measurement distribution

Models of TOJs typically assume independent sensory channels for each modality, with the measurement error of arrival latency for each channel being either Gaussian (*Schneider and Bavelier, 2003*; *Sternberg and Knoll, 1973*) or exponentially distributed (*García-Pérez and Alcalá-Quintana, 2012*; *Petrini et al., 2020*).

If arrival latencies are corrupted by Gaussian-distributed noise, the measurements of difference between auditory and visual arrival latencies $m$ also have a Gaussian distribution,

$$P(m|s) = \mathcal{N}(s + \beta, \sigma_m^2), \tag{S1}$$

where $s$ is the physical SOA, $\beta$ is the audiovisual bias, and $\sigma_m^2$ is the sum of the variabilities of auditory and visual latency. This model has a conceptual problem: the continuous Gaussian distribution allows for a non-zero probability of negative arrival latencies, but the auditory or visual signal cannot be registered in the brain before the physical stimulus occurs.

The exponential distribution of arrival latencies avoids this problem. As in the main text, the measurement of audiovisual latency $m$ has a double-exponential distribution (*Equation 1*). For both models, the order judgment is then made by comparing the arrival-latency difference and decision criteria.

### Bias-shift vs. criterion-shift model

Audiovisual temporal recalibration can result from a change in the audiovisual bias or decision criteria (*Yarrow et al., 2011a*; *Yarrow et al., 2023*). The asymmetry observed in audiovisual temporal recalibration can also be modeled by allowing for asymmetric decision criteria by expanding the simultaneity window on one side (*O'Donohue et al., 2022*; *Rohde et al., 2014*; *Yarrow et al., 2011a*). For each model, we assume that either the bias or criterion parameter changes after the recalibration phase, and is restored to a default value before the pre-test of a subsequent session. We fit the TOJ responses jointly across pre- and post-tests and all nine sessions. These TOJ models include a bias or a criterion shift as a session-dependent free parameter and keep the other parameters constant across sessions.

#### Specification of the bias-shift model

In the decision stage, the observer compares the measured SOA with a simultaneity temporal window bounded by symmetric criteria $\pm c$. Including a lapse rate $\lambda$, the psychometric functions are

$$\Psi_V(s) = \frac{\lambda}{3} + (1 - \lambda)p(m > c|s),$$

$$\Psi_A(s) = \frac{\lambda}{3} + (1 - \lambda)p(m < -c|s), \tag{S2}$$

$$\Psi_S(s) = 1 - \Psi_V(s) - \Psi_A(s).$$

where $P(m|s)$ is a Gaussian or double-exponential distribution.

In the pre-test, the parameters and the resulting predicted psychometric functions are the same across sessions. In the post-test, there is a free parameter for the shift of audiovisual bias $\beta$ after exposure to the adapter SOA within that session. Therefore, there are 9 free bias parameters, $\beta_{\text{post},j}$, one for each of the 9 sessions $j$. In sum, the parameter set $\Theta$ includes either 14 free parameters,

$\Theta = \{\tau_A, \tau_V, c, \lambda, \beta_{\text{pre}}, \{\beta_{\text{post},j}\}\}$, using a double-exponential distribution, or 13 free parameters, $\Theta = \{\sigma_m, c, \lambda, \beta_{\text{pre}}, \{\beta_{\text{post},j}\}\}$, for the Gaussian measurement distribution.

## Specification of the criterion-shift model

In the criterion-shift model, the simultaneity window is defined by a lower criterion $c_l$ and an upper criterion $c_u$, which can be asymmetric. The measurement distribution is compared with the simultaneity temporal window to obtain the psychometric functions for the three responses. The psychometric functions are the same as *Equation S2* except that we allow for an expansion of the simultaneity window in the post-test phase, thus replacing $c$ with $c_h$ in the definition of $\Psi_V$ and $-c$ with $-c_l$ in the definition of $\Psi_A$.

In the exposure phase, if the adapter SOA is inside the simultaneity window, no recalibration occurs. On the other hand, if it is outside the simultaneity window, recalibration occurs by shifting outward the criterion that is closer to the adapter SOA, i.e., expanding the range considered simultaneous on that side. The other side of the criterion remains unchanged. We assume that in a given session only the criterion on the same side as the adapter is ever changed. Thus, each session is characterized by a free parameter for how much that criterion is shifted ($\Delta_{c,j}$ for session $j$), so that either $c_h$ is updated to $c_h + \Delta_{c,j}$ or $c_l$ is updated to $c_l - \Delta_{c,j}$. All pre-test sessions share all parameters (and predicted psychometric functions). Taken together, the parameter set $\Theta$ includes either 14 free parameters, $\Theta = \{\tau_A, \tau_V, \lambda, c_l, c_u, \{\Delta_{c,j}\}\}$, for the double-exponential model, or 13 free parameters for the Gaussian model: $\Theta = \{\sigma_m, \lambda, c_l, c_u, \{\Delta_{c,j}\}\}$.

## Model log-likelihood

We fit each model $M$ separately using the same variational Bayesian Monte Carlo procedure. The TOJ data from the pre- and post-test phases of all sessions were fit jointly. Note that each model does not simulate recalibration, but fits session-dependent free parameters instead.

The parameter set $\Theta$ is specified above for each model, and $X$ is the dataset containing each type of response from $I$ phases, $J$ sessions, and $K$ trials ($I = \{\text{pre}, \text{post}\}, J = 9, K = 300$). We maximized the log-likelihood of model parameters given the data,

$$\log p(X|M, \Theta) = \sum_{i \in \{\text{pre,post}\}} \sum_{j=1}^{J} \sum_{k=1}^{K} \Big( r_{ijk}^A \log \Psi_{A,ij}(s_{ijk}) + \\ r_{ijk}^V \log \Psi_{V,ij}(s_{ijk}) + \\ r_{ijk}^S \log \Psi_{S,ij}(s_{ijk}) \Big), \tag{S3}$$

where the $r_{ijk}$ are indicator variables that are equal to one when the corresponding key was pressed, and zero otherwise. The psychometric functions (e.g., $\Psi_{A,ij}$) for the pre-test are the same across sessions and differ for the post-test across sessions due to recalibration.

## Model comparison

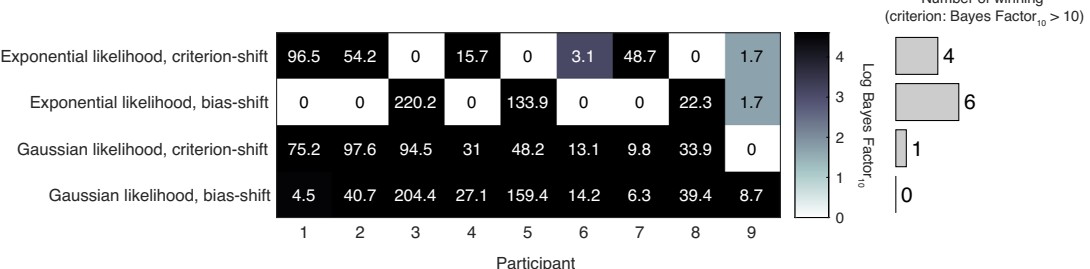

**Appendix 1—figure 1.** Model comparison results of the atheoretical models. The model with the largest log model evidence is determined to be the best-fitting model for each participant. The table displays the log Bayes factor of each model relative to the best-fitting model for each participant. Thus, the best-fitting model has a

*Appendix 1—figure 1 continued*
relative log Bayes factor of zero. Larger values for other models indicate stronger support for the best-fitting model. The vertical histogram shows the number of participants for which each model was the best fit, based on the criterion that a model is rejected if its relative log Bayes factor is greater than log(10) = 2.3.

Model comparison based on model evidence found that the bias-shift model with exponential measurement distribution captured the data of most participants. We continued to use this model as the atheoretical model as the baseline to compare with other recalibration models in the main text.

## Appendix 2

### Individual-level empirical data of the recalibration effect

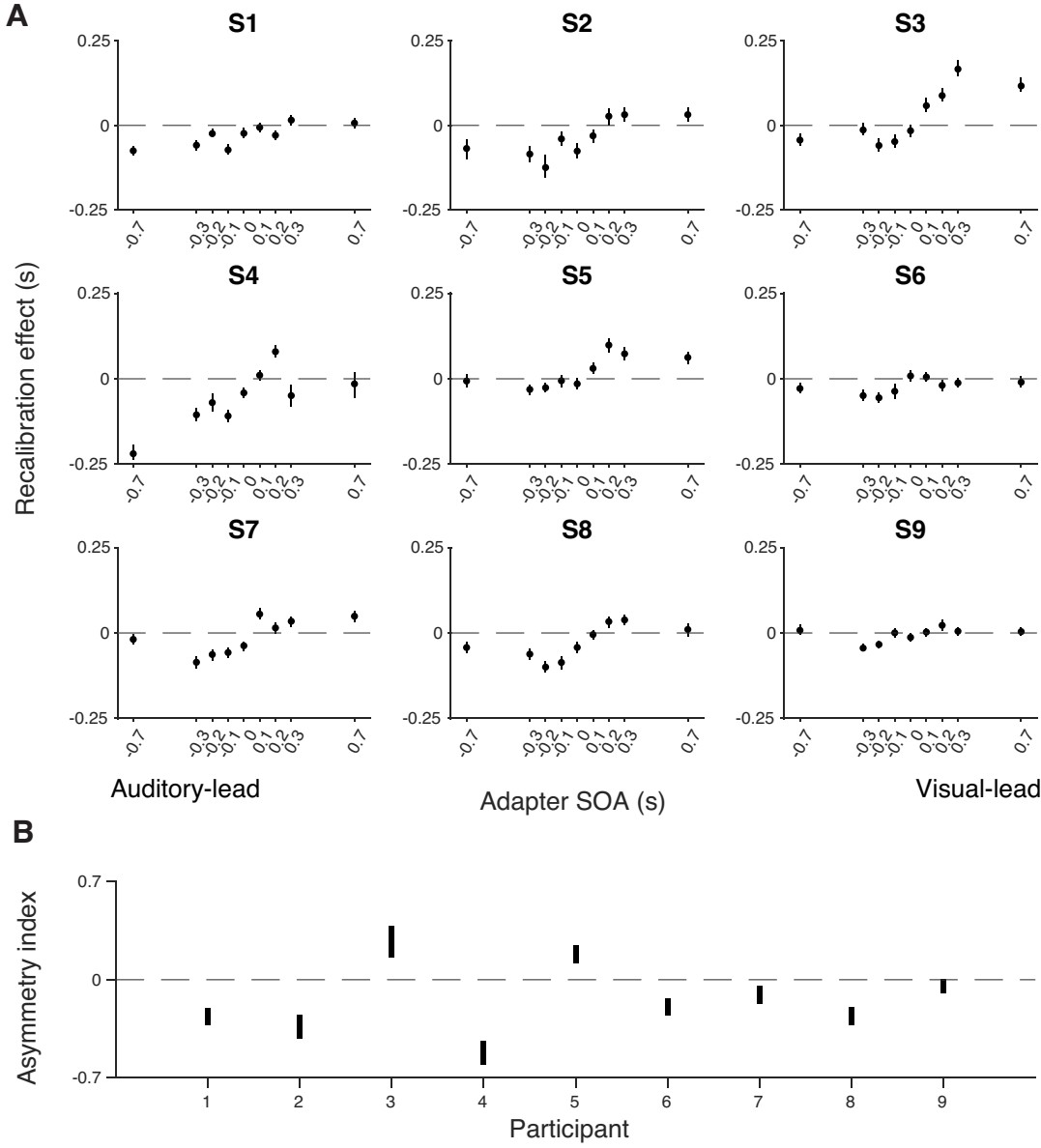

**Appendix 2—figure 1.** Individual-level recalibration effect. (**A**) Individual recalibration effect estimated from the atheoretical model that assumes a bias shift and double-exponential measurement distribution. Error bars: 95% confidence interval of recalibration amount from 1000 bootstrapped datasets for that participant. (**B**) The 95% confidence interval of the asymmetry index (i.e. summed recalibration amount across adapter SOAs) from 1000 bootstrapped datasets for each participant. The confidence interval for all except the last participant excluded zero (CI of S9: [–0.093, 0.006]), suggesting a general asymmetry in recalibration.

# Appendix 3

## Derivation of the double-exponential distribution of arrival-latency difference

In this section, we derive the asymmetric double-exponential distribution of measured SOA that results from exponential arrival-time distributions (*Equation 1* in the main text). The stimulus onsets occur at times $s_A$ and $s_V$. Each modality has a fixed delay until detection ($\beta_A$ and $\beta_V$). The probability distribution of further delays is an exponential distribution with time constant $\tau_A$ or $\tau_V$. Thus, the arrival time for auditory detection is $T_A = s_A + \beta_A + D_A$, where the random delay $D_A$ is distributed as $f_{D_A}(d_A) = \frac{e^{-d_A/\tau_A}}{\tau_A}$, and similarly for the visual stimulus. We are interested in measurements of SOA, i.e., the distribution of $D = T_A - T_V = (s_A + \beta_A) - (s_V + \beta_V) + D_A - D_V$. We now derive the distribution for the random variable $D' = D_A - D_V$.

We assume that visual and auditory delays ($D_A$ and $D_V$) are independent. Thus, to compute the density $f_{D'}(d)$ for a particular relative delay $d$, we integrate over all pairs of delays $d_V$ and $d_A = d_V + d$ that correspond to that relative delay such that both individual delays are possible (i.e. both are non-negative). We treat the cases $d \geq 0$ and $d \leq 0$ separately.

**Case 1**: $d \geq 0$

$$f_D(d) = \int_0^\infty f_{D_V}(d_V) f_{D_A}(d_V + d) \, dd_V.$$

Here, the lower limit on the integral is zero so that both delays are non-negative. Substituting the PDFs:

$$f_D(d) = \int_0^\infty \tau_V^{-1} e^{-d_V/\tau_V} \cdot \tau_A^{-1} e^{-(d_V + d)/\tau_A} \, dd_V$$

$$= \tau_V^{-1} \tau_A^{-1} e^{-d/\tau_A} \int_0^\infty e^{-d_V/\tau_V - d_V/\tau_A} \, dd_V$$

$$= \tau_V^{-1} \tau_A^{-1} e^{-d/\tau_A} \int_0^\infty e^{-(\tau_V + \tau_A)d_V/\tau_V \tau_A} \, dd_V \tag{S4}$$

$$= \tau_V^{-1} \tau_A^{-1} e^{-d/\tau_A} (\tau_V \tau_A)/(\tau_V + \tau_A)$$

$$= \frac{e^{-d/\tau_A}}{\tau_V + \tau_A}.$$

**Case 2**: $d \leq 0$

$$f_D(d) = \int_{-d}^\infty f_{D_V'}(d_V) f_{D_A'}(d_V + d) \, dd_V.$$

Here, the lower limit on the integral is $-d$ (which is a positive value) so that both delays are non-negative. Substituting the PDFs:

$$f_D(d) = \int_{-d}^{\infty} \tau_V^{-1} e^{-d_V/\tau_V} \cdot \tau_A^{-1} e^{-(d_V+d)/\tau_A} \, dd_V$$

$$= \tau_V^{-1} \tau_A^{-1} e^{-d/\tau_A} \int_{-d}^{\infty} e^{-d_V/\tau_V - d_V/\tau_A} \, dd_V$$

$$= \tau_V^{-1} \tau_A^{-1} e^{-d/\tau_A} \int_{-d}^{\infty} e^{-(\tau_V+\tau_A)d_V/\tau_V\tau_A} \, dd_V \qquad \text{(S5)}$$

$$= \tau_V^{-1} \tau_A^{-1} e^{-d/\tau_A} (\tau_V\tau_A)/(\tau_V + \tau_A) e^{(\tau_V+\tau_A)d/\tau_V\tau_A}$$

$$= e^{-d/\tau_A} e^{(1/\tau_V + 1/\tau_A)d} / (\tau_V + \tau_A)$$

$$= \frac{e^{d/\tau_V}}{\tau_V + \tau_A}.$$

Combining the two cases, the PDF of $D$ is

$$\mathrm{f}_D(d) = \begin{cases} \dfrac{e^{d/\tau_V}}{\tau_V + \tau_A}, & \text{if } d \leq 0, \\ \dfrac{e^{-d/\tau_A}}{\tau_V + \tau_A}, & \text{if } d \geq 0. \end{cases} \qquad \text{(S6)}$$

The difference of the random delays follows a double exponential distribution centered at zero, with different decay rates on either side of zero determined by $\tau_A$ and $\tau_V$. Adding in the fixed relative delays $((s_A + \beta_A) - (s_V + \beta_V))$, we arrive at *Equation 1* in the main text.

# Appendix 4

## Individual-level model comparison of the recalibration models

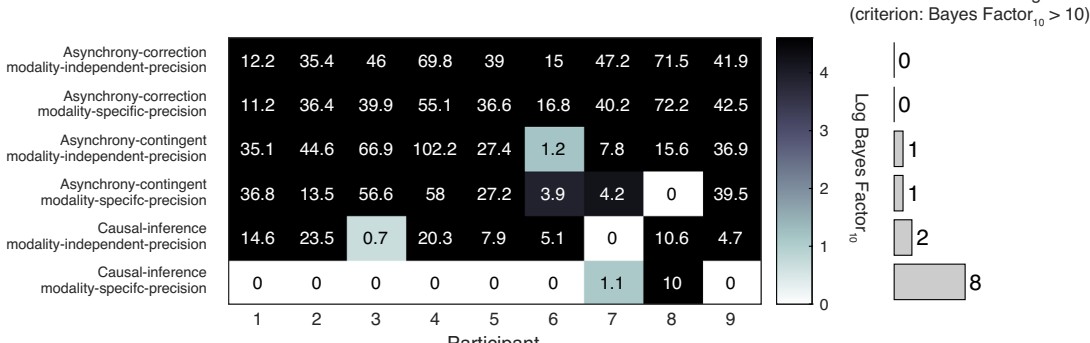

**Appendix 4—figure 1.** Model comparison results of the recalibration models. The table displays log Bayes factor relative to the best-fitting model for each participant. Thus, the best-fitting model has a relative log Bayes factor of 0. Larger values for other models indicate stronger support for the best-fitting model. The vertical histogram shows the number of participants for which each model was the best fit, based on the criterion that a model is rejected if the log Bayes factor relative to the best-fitting model is greater than log(10) = 2.3.

## Appendix 5

### Individual-level model predictions of recalibration effects

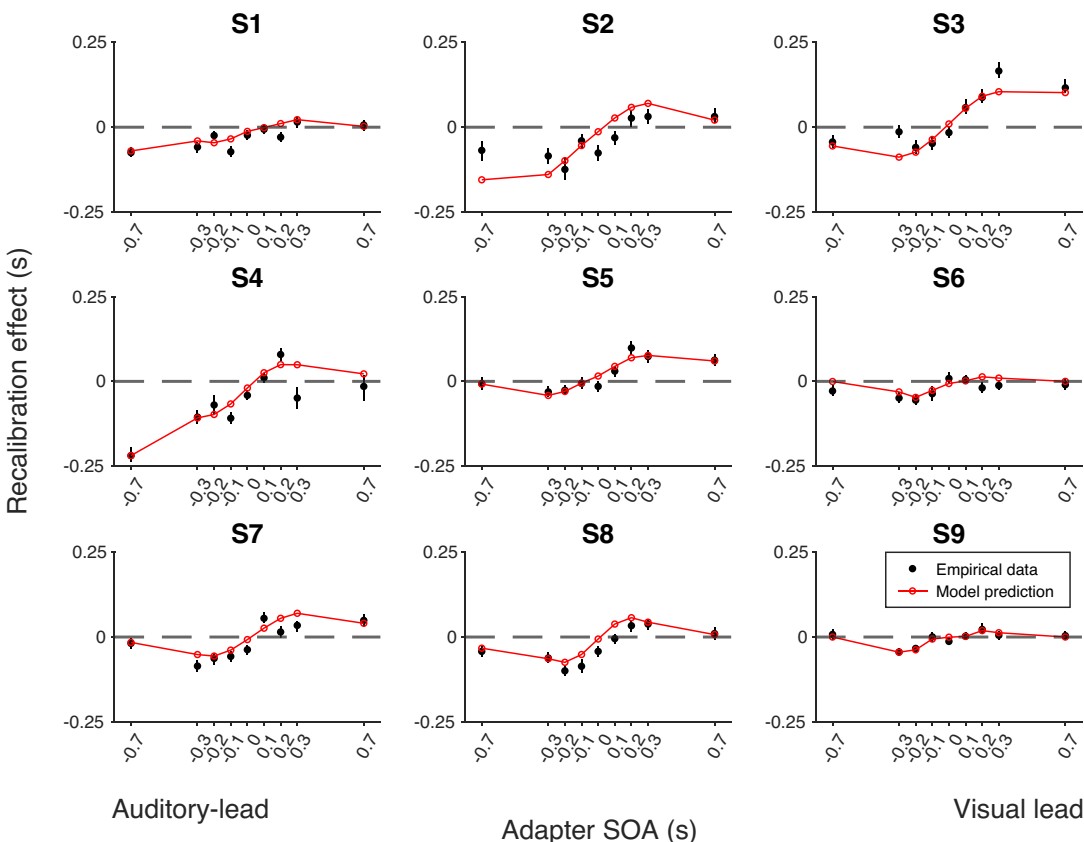

**Appendix 5—figure 1.** Empirical data for the recalibration effect for each participant and model predictions of the causal-inference recalibration model with modality-specific precision. Each panel represents a single participant. Black dots: empirical estimates from the best atheoretical model (the bias-shift model with exponential likelihood). Error bars: the 95% bootstrapped confidence intervals. Red lines: predictions of the recalibration model from the best-fitting parameter estimates for that participant.

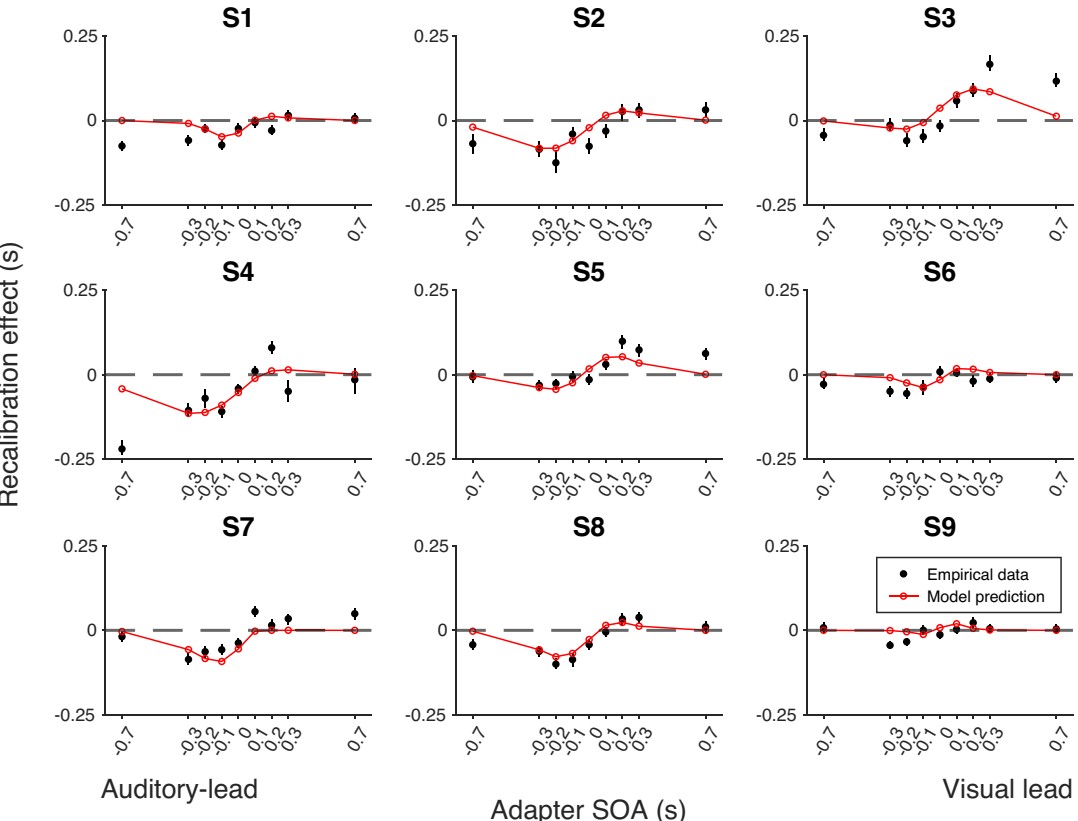

**Appendix 5—figure 2.** Empirical data of the recalibration effect for each participant and model predictions of the asynchrony-contingent recalibration model with modality-specific precision. Each panel represents a single participant. Black dots: empirical estimates from the best atheoretical model (the bias-shift model with exponential likelihood). Error bars: the 95% bootstrapped confidence intervals. Red lines: predictions of the recalibration model from the best-fitting parameter estimates for that participant.

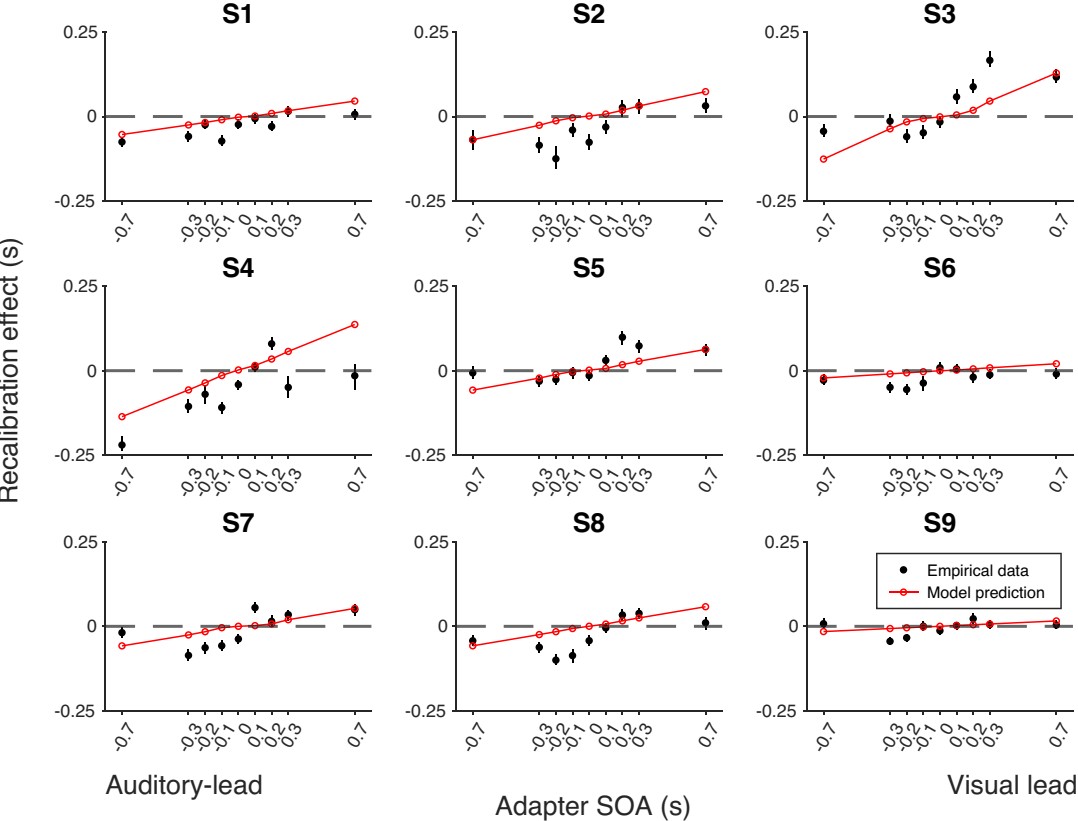

**Appendix 5—figure 3.** Empirical data of the recalibration effect for each participant and model predictions of the asynchrony-correction recalibration model with modality-specific precision. Each panel represents a single participant. Black dots: empirical estimates from the best atheoretical model (the bias-shift model with exponential likelihood). Error bars: the 95% bootstrapped confidence intervals. Red lines: predictions of the recalibration model from the best-fitting parameter estimates for that participant.

# Appendix 6

## Scrutinizing the causal-inference model with modality-specific precision

**Appendix 6—table 1.** Parameter estimates of individual participants.
All parameters except $\lambda$, $p_{\text{common}}$, and $\alpha$ are in milliseconds.

|  | $\beta_{\text{pre}}$ | $\tau_A$ | $\tau_V$ | Criterion | $\lambda$ | $p_{common}$ | $\alpha$ | $\sigma_{C=1}$ | $\sigma_{C=2}$ |
|---|---|---|---|---|---|---|---|---|---|
| S1 | −66.78 | 49.76 | 67.89 | 91.74 | 0.005 | 0.671 | 0.003 | 69.51 | 214.24 |
| S2 | 45.86 | 90.41 | 144.48 | 185.86 | 0.047 | 0.498 | 0.007 | 70.66 | 390.54 |
| S3 | 10.94 | 94.48 | 86.64 | 190.74 | 0.008 | 0.504 | 0.007 | 40.05 | 260.65 |
| S4 | −4.42 | 80.78 | 113.14 | 105.57 | 0.054 | 0.221 | 0.014 | 17.99 | 267.67 |
| S5 | 30.15 | 89.35 | 66.38 | 109.35 | 0.049 | 0.571 | 0.004 | 69.53 | 263.09 |
| S6 | −9.79 | 41.69 | 62.88 | 85.54 | 0.029 | 0.345 | 0.006 | 69.71 | 528.98 |
| S7 | −31.37 | 68.79 | 60.78 | 59.69 | 0.051 | 0.864 | 0.002 | 18.57 | 218.73 |
| S8 | 0.16 | 66.37 | 76.78 | 63.25 | 0.017 | 0.378 | 0.008 | 26.48 | 247.67 |
| S9 | 14.03 | 33.22 | 41.23 | 47.12 | 0.014 | 0.632 | 0.007 | 73.70 | 575.44 |
| Mean | −1.25 | 68.32 | 80.02 | 104.32 | 0.031 | 0.520 | 0.006 | 50.69 | 329.67 |
| SEM | 11.10 | 7.51 | 10.41 | 17.32 | 0.007 | 0.064 | 0.001 | 8.16 | 45.53 |
| Lower bound | −200 | 10 | 10 | 1 | 0 | 0 | 0 | 1 | 100 |
| Upper bound | 200 | 200 | 200 | 350 | 0.06 | 1 | 0.02 | 300 | 1000 |

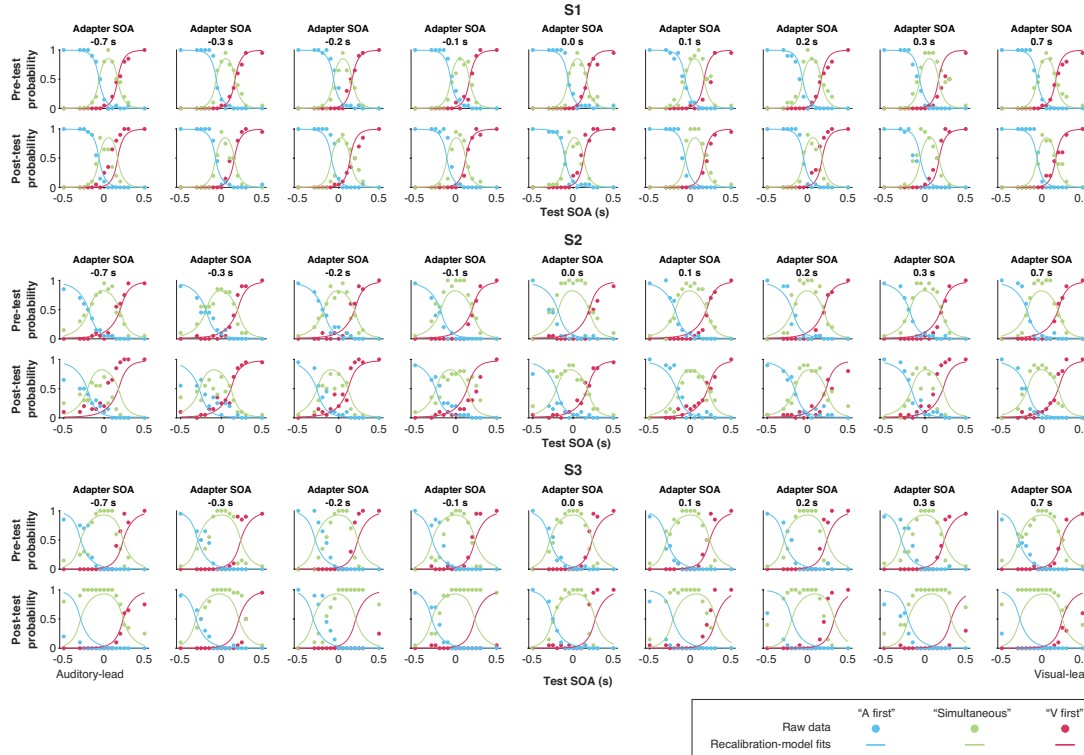

**Appendix 6—figure 1.** Data and predictions of the causal-inference, modality-specific-precision model for the temporal-order-judgment responses for each participant (S1–S3).

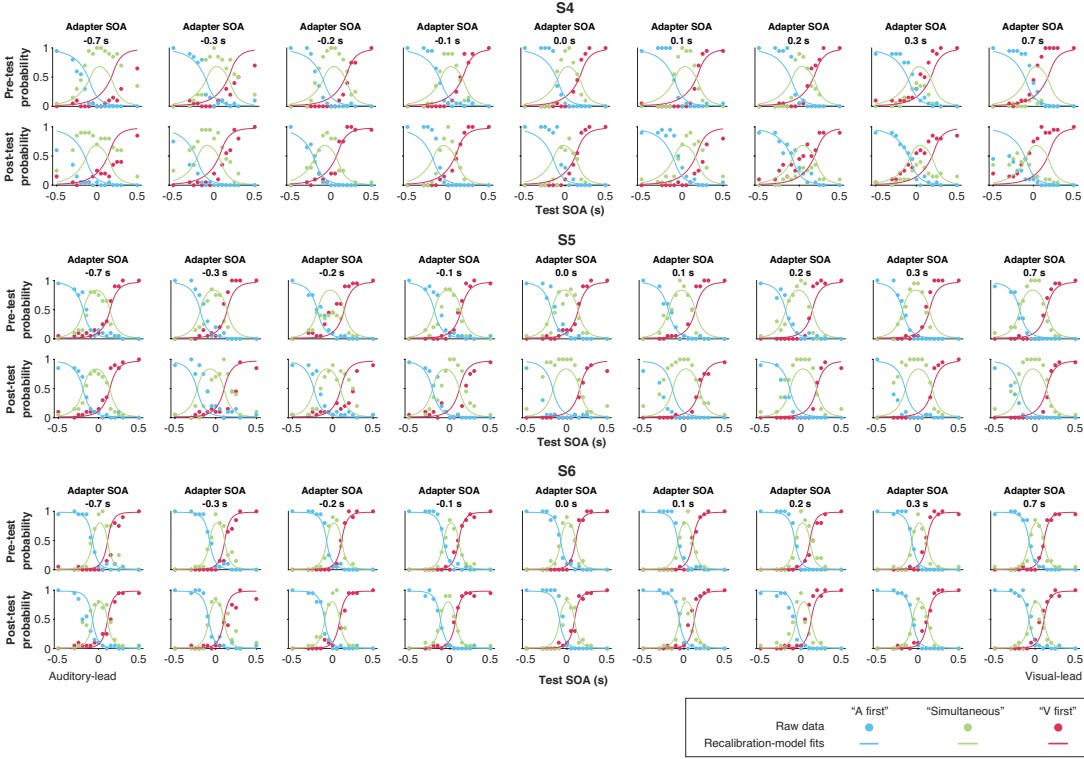

**Appendix 6—figure 2.** Data and predictions of the causal-inference, modality-specific-precision model for the temporal-order-judgment responses for each participant (S4–S6).

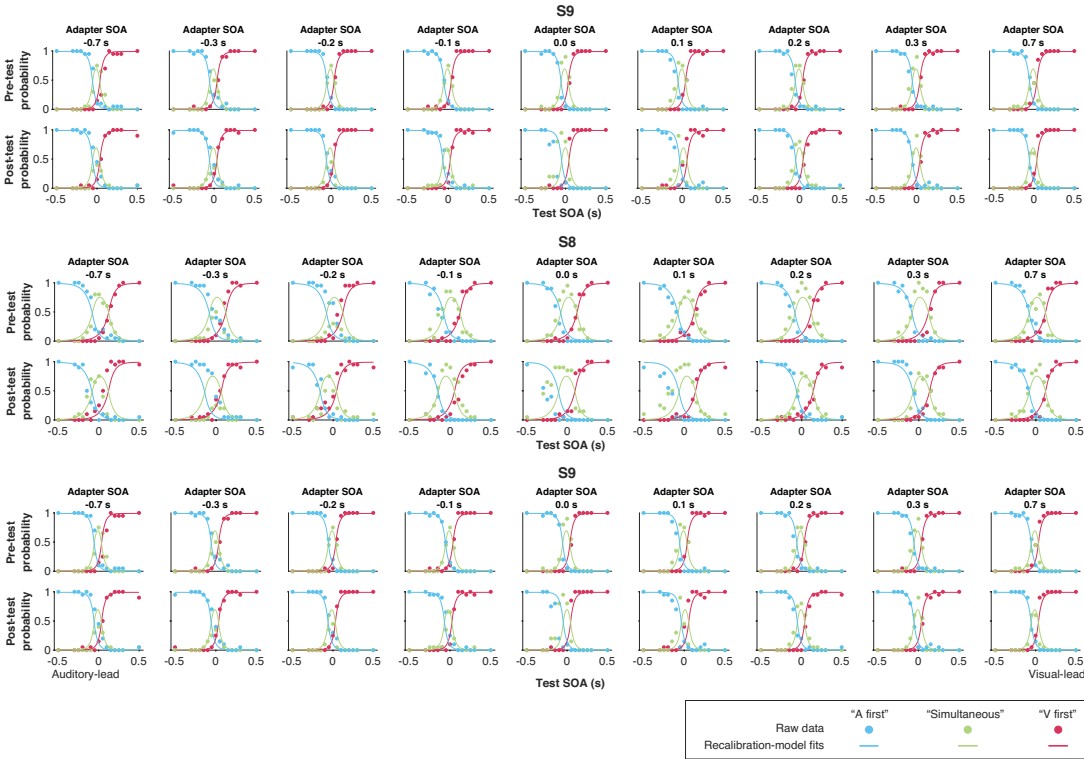

**Appendix 6—figure 3.** Data and predictions of the causal-inference, modality-specific-precision model for the temporal-order-judgment responses for each participant (S7–S9).

## Appendix 7

## Simulation of the causal-inference model

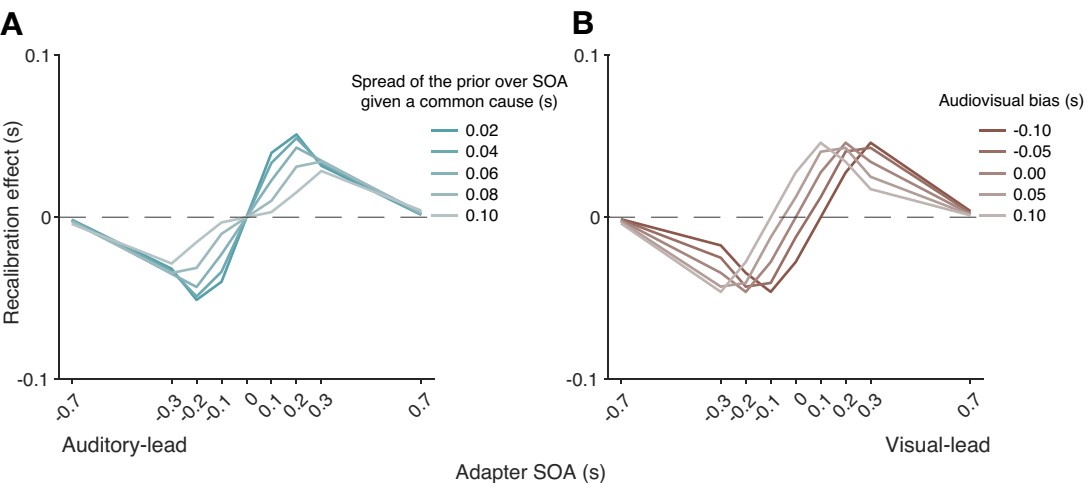

**Appendix 7—figure 1.** The effect of the spread of the prior over SOA given a common cause on recalibration. (**A**) A smaller spread increases the recalibration magnitude, but only for a small SOA range where the probability of a common cause is higher. (**B**) The effect of the audiovisual temporal bias on recalibration. Audiovisual temporal bias affects where recalibration starts in the direction of the other modality, but its effects are reduced for large adapter SOAs. Overall, temporal bias shifts the recalibration function laterally, with minimal impact on the asymmetry of recalibration effect.

## Appendix 8

## Simulation of a population-code recalibration model

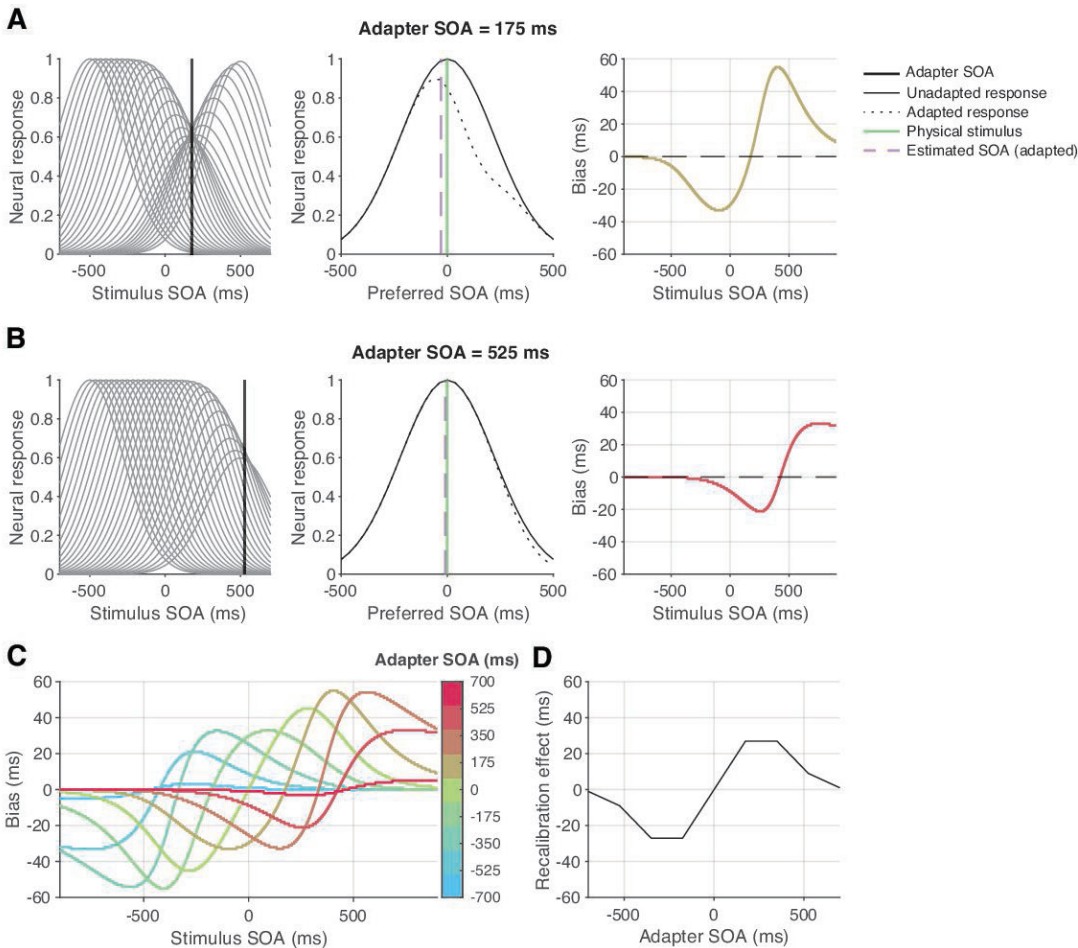

**Appendix 8—figure 1.** Simulation of a population-code model with varying adapter SOA. (**A**) Left panel: Tuning curves of a neural population adapted to a 175 ms SOA, with 29 neurons having preferred SOAs that range from –500 to 500 ms. Middle panel: Neural responses to a stimulus with 0 ms SOA before (solid) and after (dashed) adaptation to a 175 ms SOA. Green: physical 0 ms SOA. Purple: estimated SOA based on an adaptation-unaware maximum-likelihood readout of the population response after adaptation. Right panel: Bias (i.e. estimated SOA minus physical SOA) across different stimulus SOAs after adaptation to a 175 ms SOA. (**B**) The same as in (**A**) after adaptation to a 525 ms SOA. (**C**) Bias as a function of stimulus SOA for several adapter SOAs. (**D**) Recalibration effect (i.e. the points of subjective simultaneity [PSS]: the stimulus SOA leading to an estimate of a 0 ms SOA) assuming an unbiased PSS before adaptation.

We simulated the population-code model of audiovisual temporal recalibration (*Roach et al., 2011*; *Yarrow et al., 2015*) to examine its prediction of the recalibration effect (i.e. the shift of the PSS) for different adapter SOAs. This model proposes that a population of neurons is tuned to different SOAs, and recalibration occurs through a selective gain reduction around the adapter SOA (*Appendix 8—figure 1A*, left panel). We assume a maximum-likelihood decoder that is unaware of the adaptation, resulting in a bias: estimates are shifted away from the adapted SOA (*Appendix 8—figure 1A*, middle panel; *Jazayeri and Movshon, 2006*; *Seriès et al., 2009*). The bias (i.e. the difference between estimated and physical SOA) is nonuniform as a function of the test SOA (*Appendix 8—figure 1A*, right panel).

When the adapter SOA is close to or exceeds the range of preferred SOAs across the neural population, it elicits less adaptation across neurons (*Appendix 8—figure 1B*, left panel) and results in a smaller change in population response (*Appendix 8—figure 1B*, middle panel). Consequently,

the bias is significantly reduced as the adapter SOA increases (*Appendix 8—figure 1B*, right panel).

We simulated the bias in estimation as a function of stimulus SOA for this neural population when adapted to a range of different adapter SOAs (*Appendix 8—figure 1C*). These adapter SOAs produce a consistent nonuniform bias across stimulus SOAs. The overall magnitude of the bias is largest for small adapter SOAs.

Finally, we determined the recalibration effect by calculating the PSS after adaptation for each adapter SOA, assuming an unbiased PSS before adaptation (*Appendix 8—figure 1D*). For each adapter SOA we determined the stimulus SOA that, after adaptation, resulted in an SOA estimate of 0 ms, i.e., the stimulus SOA plus the bias is zero. The results show that the population-code model can predict nonlinearity in recalibration because the strength of adaptation decreases as the adapter SOA approaches the bounds of the range of preferred SOAs across the neural population. However, it cannot account for the asymmetry of recalibration observed in the empirical data, under the assumption that neurons representing relative timing, a supramodal attribute, have similar selectivity in this model regardless of which modality leads.

# Appendix 9

## Exclusion of an outlier participant

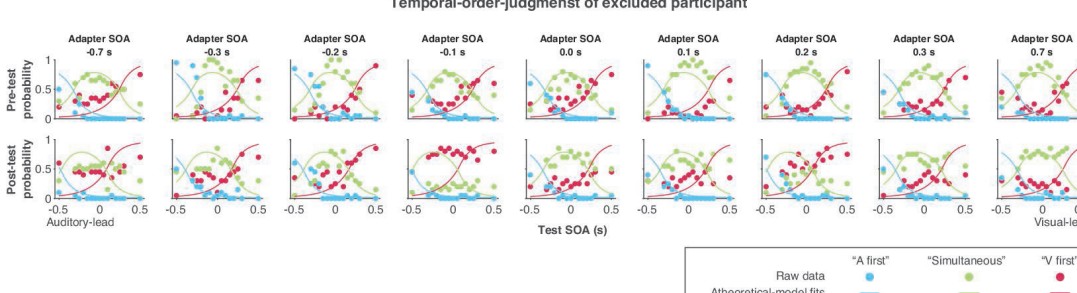

**Appendix 9—figure 1.** Temporal-order judgments from the outlier participant. We excluded this participant's data because the probabilities of the three responses were similar across test stimulus-onset asynchronies (SOAs). These responses were too noisy to constrain the psychometric function.

## Appendix 10

### Performance in the oddball-detection task

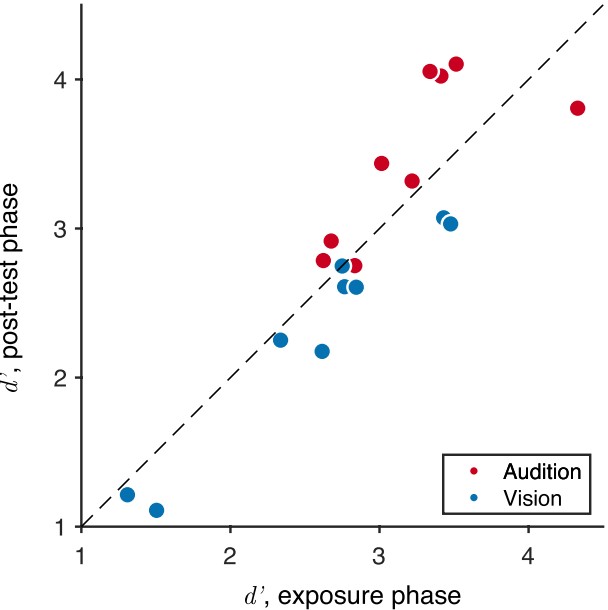

**Appendix 10—figure 1.** Oddball-detection performance in the exposure and post-test phases. Each dot represents a participant. In each phase, we combined responses across sessions and computed hit rates and false-alarm rates for auditory and visual oddballs. For each modality, hit and false-alarm rates were the probability of reporting an oddball when an oddball was presented and when it was absent, respectively. The resulting $d'$ values for each participant confirm that they engaged in the task and attended to both modalities during the exposure phase. In general, participants showed a higher $d'$ for auditory oddballs compared to visual ones, despite our attempt to equate performance levels. Furthermore, the $d'$ remained consistent from the exposure to post-test phases, indicating similar performance in the top-up trials to performance during the exposure phase.

## Appendix 11

### Model recovery of recalibration models

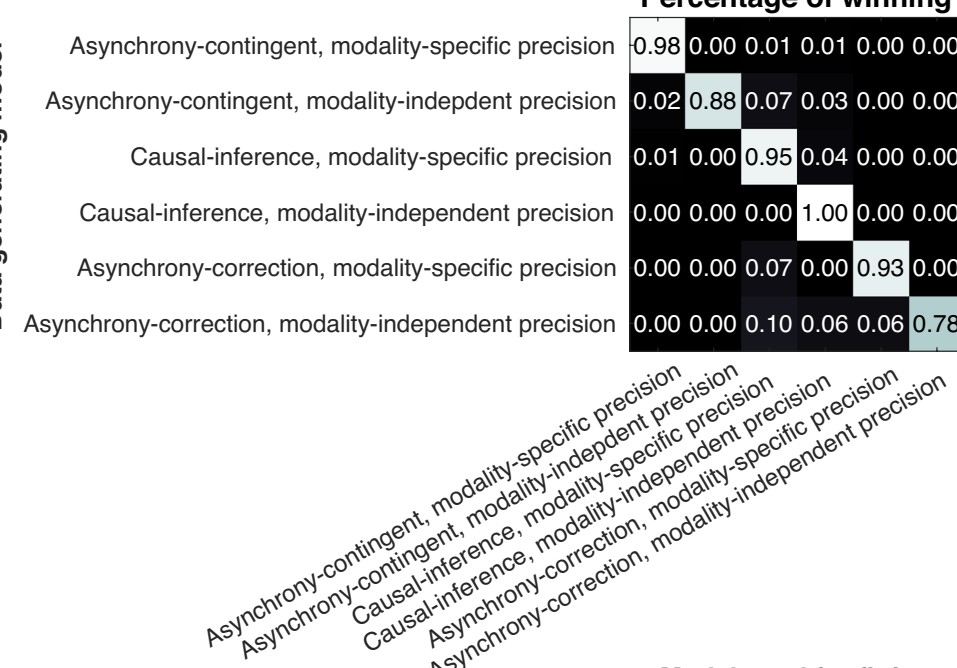

**Appendix 11—figure 1.** Confusion matrix resulting from the model-recovery simulations. Each cell indicates the proportion of the 100 simulations for each model generating the data that were best fit by each of the six models (i.e. the values in each row sum to one).

We conducted a model-recovery analysis for the six models described in the main text. In the main text, each model was fit to the data of each participant multiple times and the best-fitting parameter set for that model was selected. This yielded one parameter set for each participant and model. For each model, we randomly selected 100 parameter sets. Each parameter was randomly drawn from either a Gaussian (for unbounded parameters) or log-Gaussian (for parameters restricted to be positive) with mean and SD equal to that of the best-fit parameter sets across participants for that model. We then simulated datasets for each model based on each of the 100 parameter sets. All six models were fit to each simulated dataset.

We used the same model-comparison procedure as in the main text. All asynchrony-contingent and causal-inference models have satisfactory recovery results, suggesting that these models are identifiable. In addition, the data generated by the asynchrony-correction model were sometimes best fit by causal-inference models but not by the asynchrony-contingent models. This is reasonable since the asynchrony-correction model is similar to the causal-inference model with the value of $p_{common}$ set to one.

## Appendix 12

### Model recovery of variants of the causal-inference recalibration models

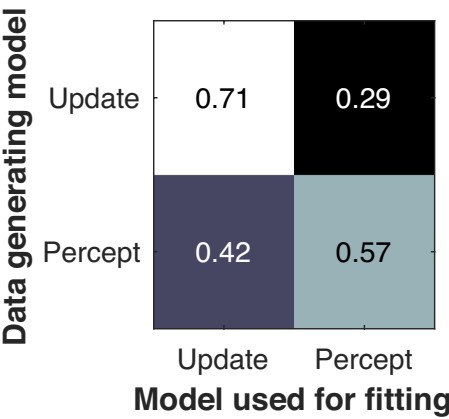

**Appendix 12—figure 1.** Confusion matrix resulting from the model-recovery simulations. Update model: the alternative model assuming that causal inference modulates recalibration at the bias-update phase. Percept model: the model described in the main text that assumes that causal inference modulates recalibration at the perceptual stage.

We compared the causal-inference recalibration model in the main text to a variant in which causal inference modulates recalibration in the update phase. Specifically, the learning rate is modulated by the posterior probability of a common cause, i.e., the probability that the audiovisual measurements arose from the same source. The bias update rule (*Equation 2*) becomes

$$\Delta_{\beta,i+1} = \Delta_{\beta,i} - P(C = 1 | m_i)\alpha m_i. \tag{S7}$$

We performed a model-recovery analysis comparing these two models. 120 datasets were simulated for each model using the best-fitting parameters from the fit to the raw data and we fit both models to each simulated dataset. The confusion matrix summarizes the probability that the generating model is the best-fitting model. Clearly, the results indicate that these two models are not distinguishable given the amount of data we collected in the main experiment.

## Appendix 13

### Parameter recovery of the causal-inference, modality-specific-precision model

We conducted a parameter recovery analysis of the winning model by simulating 100 datasets. Parameter values were sampled using the group mean and standard deviation across participants of the best-fitting estimates from model fits of the data. Each dataset was fit with the same causal-inference model with modality-specific precision. Key parameters, including audiovisual bias $\beta_{\text{pre}}$, amount of auditory latency noise $\tau_A$, amount of visual latency noise $\tau_V$, criterion, and lapse rate $\lambda$ showed satisfactory recovery performance. The less accurate recovery of $p_{common}$ is likely due to a tradeoff with the learning rate $\alpha$, as both parameters influence the update of the audiovisual bias. The less accurate recovery of the widths of the SOA for a common cause ($\sigma_{C=1}$) and separate causes ($\sigma_{C=2}$) is acceptable, as their function can still be fulfilled within the respective range.

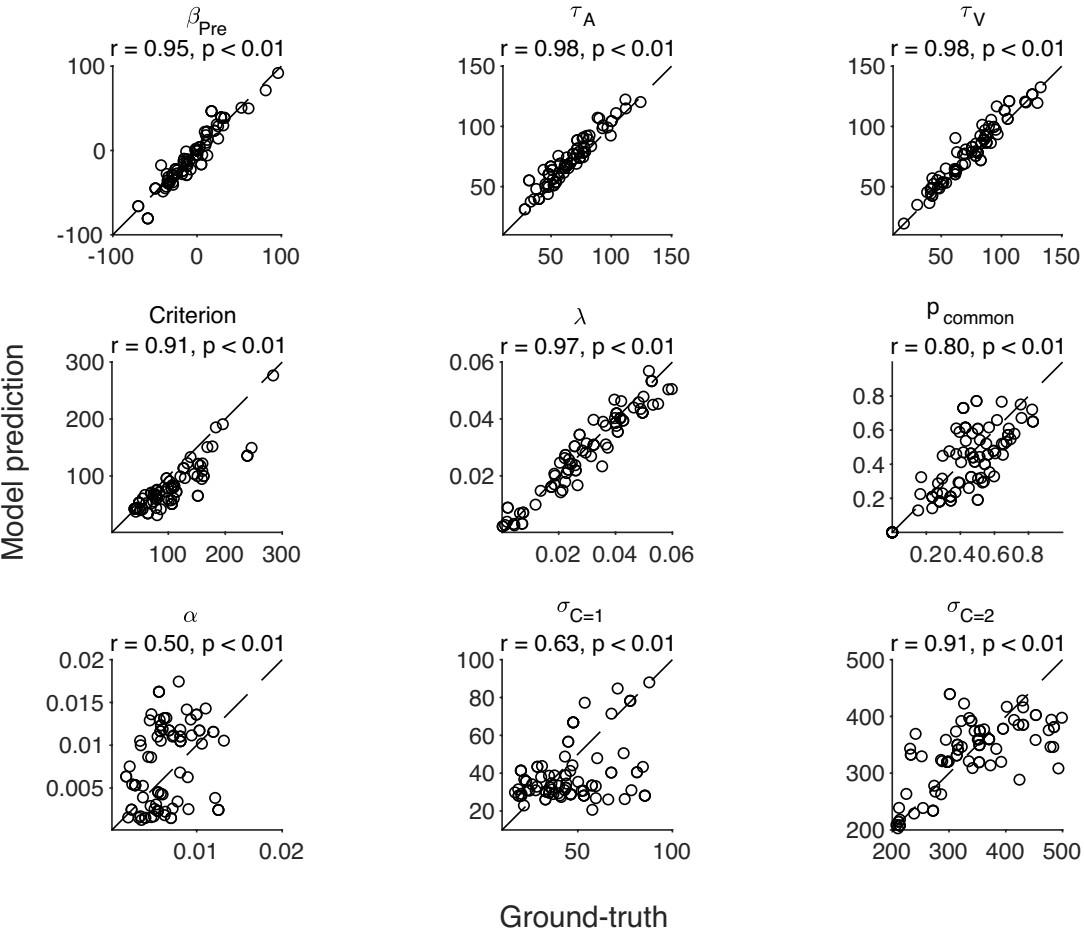

**Appendix 13—figure 1.** Predicted value vs. simulated value of each parameter. Each dot represents 1 out of 100 simulated datasets. The identity line marks perfect recovery performance.

