## [Editor Report · eLife Assessment]

In this **valuable** study, Li et al., set out to understand the mechanisms of audiovisual temporal recalibration - the brain's ability to adjust to the latency differences that emerge due to different (distance-dependent) transduction latencies of auditory and visual signals - through psychophysical measurements and modeling. The analysis and specification of a formal model for this process provide **convincing** evidence to support a role for causal inference in recalibration.

---

## [Referee Report · Reviewer #1 (Public review)]

This study asks whether the phenomenon of crossmodal temporal recalibration, i.e. the adjustment of time perception by consistent temporal mismatches across the senses, can be explained by the concept of multisensory causal inference. In particular they ask whether the explanation offered by causal inference better explains temporal recalibration better than a model assuming that crossmodal stimuli are always integrated, regardless of how discrepant they are.

The study is motivated by previous work in the spatial domain, where it has been shown consistently across studies that the use of crossmodal spatial information is explained by the concept of multisensory causal inference. It is also motivated by the observation that the behavioral data showcasing temporal recalibration feature nonlinearities that, by their nature, cannot be explained by a fixed integration model (sometimes also called mandatory fusion).

To probe this the authors implemented a sophisticated experiment that probed temporal recalibration in several sessions. They then fit the data using the two classes of candidate models and rely model criteria to provide evidence for their conclusion. The study is sophisticated, conceptually and technically state-of-the-art and theoretically grounded. The data clearly support the authors conclusions.

I find the conceptual advance somewhat limited. First, by design the fixed integration model cannot explain data with a nonlinear dependency on multisensory discrepancy, as already explained in many studies on spatial multisensory perception. Hence, it is not surprising that the causal inference model better fits the data. Second, and again similar to studies on spatial paradigms, the causal inference model fails to predict the behavioral data for large discrepancies. The model predictions in Figure 5 show the (expected) vanishing recalibration for large delta, while the behavioral data don't' decay to zero. Either the range of tested SOAs is too small to show that both the model and data converge to the same vanishing effect at large SOAs, or the model's formula is not the best for explaining the data. Again, the studies using spatial paradigms have the same problem, but in my view this poses the most interesting question here.

In my view there is nothing generally wrong with the study, it does extend the 'known' to another type of paradigm. However, it covers little new ground on the conceptual side.

On that note, the small sample size of n=10 is likely not an issue, but still it is on the very low end for this type of study.

Comments on revision:

The revision has addressed most of these points and makes for a much stronger contribution. The issue of sample size remains.

---

## [Referee Report · Reviewer #2 (Public review)]

Summary:

Li et al.'s goal is to understand the mechanisms of audiovisual temporal recalibration. This is an interesting challenge that the brain readily solves in order to compensate for real-world latency differences in the time of arrival of audio/visual signals. To do this they perform a 3-phase recalibration experiment on 9 observers that involves a temporal order judgment (TOJ) pretest and posttest (in which observers are required to judge whether an auditory and visual stimulus were coincident, auditory leading or visual leading) and a conditioning phase in which participants are exposed to a sequence of AV stimuli with a particular temporal disparity. Participants are required to monitor both streams of information for infrequent oddballs, before being tested again in the TOJ, although this time there are 3 conditioning trials for every 1 TOJ trial. Like many previous studies, they demonstrate that conditioning stimuli shift the point of subjective simultaneity (pss) in the direction of the exposure sequence.

These shifts are modest - maxing out at around -50 ms for auditory leading sequences and slightly less than that for visual leading sequences. Similar effects are observed even for the longest offsets where it seems unlikely listeners would perceive the stimuli as synchronous (and therefore under a causal inference model you might intuitively expect no recalibration, and indeed simulations in Figure 5 seem to predict exactly that which isn't what most of their human observers did). Overall I think their data contribute evidence that a causal inference step is likely included within the process of recalibration.

Strengths:

The manuscript performs comprehensive testing over 9 days and 100s of trials and accompanies this with mathematical models to explain the data. The paper is reasonably clearly written and the data appear to support the conclusions.

Comments on revision:

In the revised manuscript the authors incorporate an alternative model (the asynchrony contingent model), and demonstrate that the causal inference model still out performs this. They provide additional analysis with Bayes factors to perform model comparisons, and provide significant individual subject data in the supplementary materials. Overall they have addressed most of the key points that my original review raised, including a demonstration of the conditions under which recalibration effects do not delay to zero over long delays. The number of subjects remains rather low, but at least we can now appreciate the heterogeneity within them. I still have some reservations about the magnitude of the conceptual advance that this study makes.

---

## [Referee Report · Reviewer #3 (Public review)]

Summary:

Li et al. describe an audiovisual temporal recalibration experiment in which participants perform baseline sessions of ternary order judgments about audiovisual stimulus pairs with various stimulus-onset asynchronies (SOAs). These are followed by adaptation at several adapting SOAs (each on a different day), followed by post-adaptation sessions to assess changes in psychometric functions. The key novelty is the formal specification and application/fit of a causal-inference model for the perception of relative timing, providing simulated predictions for the complete set of psychometric functions both pre and post adaptation.

Strengths:

(1) Formal models are preferable to vague theoretical statements about a process, and prior to this work, certain accounts of temporal recalibration (specifically those that do not rely on a population code) had only qualitative theoretical statements to explain how/why the magnitude of recalibration changes non-linearly with the stimulus-onset asynchrony of the adaptor.

(2) The experiment is appropriate, the methods are well described, and the average model prediction is a good match to the average data (Figure 4). Conclusions are supported by the data and modelling.

(3) The work should be impactful. There seems a good chance that this will become the go-to modelling framework for those exploring non population-code accounts of temporal recalibration (or comparing them with population-code accounts).

(4) Key issues for the generality of the model, such as recalibration asymmetries reported by other authors that are inconsistent with those reported here, are thoughtfully discussed.

Weaknesses:

(1) Models are not compared using a gold-standard measure such as leave-one-out cross validation. However, this is legitimate given lengthy model fitting times, and a sensible approximation is presented.

(2) The model misses in a systematic way for the psychometric functions of some participants/conditions. In addition to misses relating to occasional failures to estimate the magnitude of recalibration, some of the misses are because all functions are only permitted to shift in central tendency (whereas some participants show changes better characterized at one or both decision criteria). Given the fact that the modelling in general embraces individual differences, it might have been worth allowing different kinds of change for different participants. However, this is not really critical for the central concern (changes in the magnitude of recalibration for different adaptors) and there is a limit to how much can be done along these lines without making the model too flexible to test.

(3) As a minor point, the model relies on simulation, which may limit its take-up/application by others in the field (although open access code will be provided).

---

## [Author Response]

The following is the authors’ response to the original reviews.

**Public Reviews:**

**Reviewer #1 (Public Review):**
This study asks whether the phenomenon of crossmodal temporal recalibration, i.e. the adjustment of time perception by consistent temporal mismatches across the senses, can be explained by the concept of multisensory causal inference. In particular, they ask whether the explanation offered by causal inference better explains temporal recalibration better than a model assuming that crossmodal stimuli are always integrated, regardless of how discrepant they are.The study is motivated by previous work in the spatial domain, where it has been shown consistently across studies that the use of crossmodal spatial information is explained by the concept of multisensory causal inference. It is also motivated by the observation that the behavioral data showcasing temporal recalibration feature nonlinearities that, by their nature, cannot be explained by a fixed integration model (sometimes also called mandatory fusion).To probe this the authors implemented a sophisticated experiment that probed temporal recalibration in several sessions. They then fit the data using the two classes of candidate models and rely on model criteria to provide evidence for their conclusion. The study is sophisticated, conceptually and technically state-of-the-art, and theoretically grounded. The data clearly support the authors’ conclusions.I find the conceptual advance somewhat limited. First, by design, the fixed integration model cannot explain data with a nonlinear dependency on multisensory discrepancy, as already explained in many studies on spatial multisensory perception. Hence, it is not surprising that the causal inference model better fits the data.

We have addressed this comment by including an asynchrony-contingent model, which is capable of predicting the nonlinearity of recalibration effects by employing a heuristic approximation of the causal-inference process (Fig. 3). We also updated the previous competitor model with a more reasonable asynchrony-correction model as the baseline of model comparison, which assumes recalibration aims to restore synchrony whenever the sensory measurement of SOA indicates an asynchrony. The causal-inference model outperformed both models, as indicated by model evidence (Fig. 4A). Furthermore, model predictions show that the causal-inference model more accurately captures recalibration at large SOAs at both the group (Fig. 4B) and the individual levels (Fig. S4).

Second, and again similar to studies on spatial paradigms, the causal inference model fails to predict the behavioral data for large discrepancies. The model predictions in Figure 5 show the (expected) vanishing recalibration for large delta, while the behavioral data don’t decay to zero. Either the range of tested SOAs is too small to show that both the model and data converge to the same vanishing effect at large SOAs, or the model's formula is not the best for explaining the data. Again, the studies using spatial paradigms have the same problem, but in my view, this poses the most interesting question here.

We included an additional simulation (Fig. 5B) to show that the causal-inference model can predict non-zero recalibration for long adapter SOAs, especially in observers with a high common-cause prior and low sensory precision. This ability to predict a non-zero recalibration effect even at large SOA, such as 0.7 s, is one key feature of the causal-inference model that distinguishes it from the asynchrony-contingent model.

In my view there is nothing generally wrong with the study, it does extend the 'known' to another type of paradigm. However, it covers little new ground on the conceptual side.On that note, the small sample size of n=10 is likely not an issue, but still, it is on the very low end for this type of study.

This study used a within-subject design, which included 3 phases each repeated in 9 sessions, totaling 13.5 hours per participant. This extensive data collection allows us to better constrain the model for each participant. Our conclusions are based on the different models’ ability to fit individual data.

**Reviewer #2 (Public Review):**
Summary:Li et al.’s goal is to understand the mechanisms of audiovisual temporal recalibration. This is an interesting challenge that the brain readily solves in order to compensate for real-world latency differences in the time of arrival of audio/visual signals. To do this they perform a 3-phase recalibration experiment on 9 observers that involves a temporal order judgment (TOJ) pretest and posttest (in which observers are required to judge whether an auditory and visual stimulus were coincident, auditory leading or visual leading) and a conditioning phase in which participants are exposed to a sequence of AV stimuli with a particular temporal disparity. Participants are required to monitor both streams of information for infrequent oddballs, before being tested again in the TOJ, although this time there are 3 conditioning trials for every 1 TOJ trial. Like many previous studies, they demonstrate that conditioning stimuli shift the point of subjective simultaneity (pss) in the direction of the exposure sequence.These shifts are modest - maxing out at around -50 ms for auditory leading sequences and slightly less than that for visual leading sequences. Similar effects are observed even for the longest offsets where it seems unlikely listeners would perceive the stimuli as synchronous (and therefore under a causal inference model you might intuitively expect no recalibration, and indeed simulations in Figure 5 seem to predict exactly that which isn't what most of their human observers did). Overall I think their data contribute evidence that a causal inference step is likely included within the process of recalibration.Strengths:The manuscript performs comprehensive testing over 9 days and 100s of trials and accompanies this with mathematical models to explain the data. The paper is reasonably clearly written and the data appear to support the conclusions.Weaknesses:While I believe the data contribute evidence that a causal inference step is likely included within the process of recalibration, this to my mind is not a mechanism but might be seen more as a logical checkpoint to determine whether whatever underlying neuronal mechanism actually instantiates the recalibration should be triggered.

We have addressed this comment by replacing the fixed-update model with an asynchrony-correction model, which assumes that the system first evaluates whether the measurement of SOA is asynchronous, thus indicating a need for recalibration (Fig. 3). If it does, it shifts the audiovisual bias by a proportion of the measured SOA. We additionally included an asynchrony-contingent model, which is capable of replicating the nonlinearity of recalibration effects by a heuristic approximation of the causal-inference process.

Model comparisons indicate that the causal-inference model of temporal recalibration outperforms both alternative models (Fig. 4A). Furthermore, the model predictions demonstrate that the causal-inference model more accurately captures recalibration at large SOAs at both the group level (Fig. 4B) and individual level (Fig. S4).

The authors’ causal inference model strongly predicts that there should be no recalibration for stimuli at 0.7 ms offset, yet only 3/9 participants appear to show this effect. They note that a significant difference in their design and that of others is the inclusion of longer lags, which are unlikely to originate from the same source, but don’t offer any explanation for this key difference between their data and the predictions of a causal inference model.

We added further simulations to show that the causal-inference model can predict non-zero recalibration also for longer adapter SOAs, especially in observers with a large common-cause prior (Fig. 5A) and low sensory precision (Fig. 5B). This ability to predict a non-zero recalibration effect even at longer adapter SOAs, such as 0.7 s, is a key feature of the causal-inference model that distinguishes it from the asynchrony-contingent model.

I’m also not completely convinced that the causal inference model isn’t ‘best’ simply because it has sufficient free parameters to capture the noise in the data. The tested models do not (I think) have equivalent complexity - the causal inference model fits best, but has more parameters with which to fit the data. Moreover, while it fits ‘best’, is it a good model? Figure S6 is useful in this regard but is not completely clear - are the red dots the actual data or the causal inference prediction? This suggests that it does fit the data very well, but is this based on predicting held-out data, or is it just that by having more parameters it can better capture the noise? Similarly, S7 is a potentially useful figure but it's not clear what is data and what are model predictions (what are the differences between each row for each participant; are they two different models or pre-test post-test or data and model prediction?!).I'm not an expert on the implementation of such models but my reading of the supplemental methods is that the model is fit using all the data rather than fit and tested on held-out data. This seems problematic.We recognize the risk of overfitting with the causal-inference model. We now rely on Bayesian model comparisons, which use model evidence for model selection. This method automatically incorporates a penalty for model complexity through the marginalization over the parameter space (MacKay, 2003).Our design is not suitable for cross-validation because the model-fitting process is computationally intensive and time-consuming. Each fit of the causal-inference model takes approximately 30 hours, and multiple fits with different initial starting points are required to rule out that the parameter estimates correspond to local minima.

I would have liked to have seen more individual participant data (which is currently in the supplemental materials, albeit in a not very clear manner as discussed above).

We have revised Supplementary Figures S4-S6 to show additional model predictions of the recalibration effect for individual participants, and participants’ temporal-order judgments are now shown in Supplement Figure S7. These figures confirm the better performance of the causal-inference model.

The way that S3 is described in the text (line 141) makes it sound like everyone was in the same direction, however, it is clear that 2 /9 listeners show the opposite pattern, and 2 have confidence intervals close to zero (albeit on the -ve side).

We have revised the text to clarify that the asymmetry occurs in both directions and is idiosyncratic (lines 168-171). We summarized the distribution of the individual asymmetries of the recalibration effect across visual-leading and auditory-leading adapter SOAs in Supplementary Figure S2.

**Reviewer #3 (Public Review):**
Summary:Li et al. describe an audiovisual temporal recalibration experiment in which participants perform baseline sessions of ternary order judgments about audiovisual stimulus pairs with various stimulus-onset asynchronies (SOAs). These are followed by adaptation at several adapting SOAs (each on a different day), followed by post-adaptation sessions to assess changes in psychometric functions. The key novelty is the formal specification and application/fit of a causal-inference model for the perception of relative timing, providing simulated predictions for the complete set of psychometric functions both pre and post-adaptation.Strengths:(1) Formal models are preferable to vague theoretical statements about a process, and prior to this work, certain accounts of temporal recalibration (specifically those that do not rely on a population code) had only qualitative theoretical statements to explain how/why the magnitude of recalibration changes non-linearly with the stimulus-onset asynchrony of the adapter.(2) The experiment is appropriate, the methods are well described, and the average model prediction is a fairly good match to the average data (Figure 4). Conclusions may be overstated slightly, but seem to be essentially supported by the data and modelling.(3) The work should be impactful. There seems a good chance that this will become the go-to modelling framework for those exploring non-population-code accounts of temporal recalibration (or comparing them with population-code accounts).(4) A key issue for the generality of the model, specifically in terms of recalibration asymmetries reported by other authors that are inconsistent with those reported here, is properly acknowledged in the discussion.Weaknesses:(1) The evidence for the model comes in two forms. First, two trends in the data (non-linearity and asymmetry) are illustrated, and the model is shown to be capable of delivering patterns like these. Second, the model is compared, via AIC, to three other models. However, the main comparison models are clearly not going to fit the data very well, so the fact that the new model fits better does not seem all that compelling. I would suggest that the authors consider a comparison with the atheoretical model they use to first illustrate the data (in Figure 2). This model fits all sessions but with complete freedom to move the bias around (whereas the new model constrains the way bias changes via a principled account). The atheoretical model will obviously fit better, but will have many more free parameters, so a comparison via AIC/BIC or similar should be informative

In the revised manuscript, we switched from AIC to Bayesian model selection, which approximates and compares model evidence. This method incorporates a strong penalty for model complexity through marginalization over the parameter space (MacKay, 2003).

We have addressed this comment by updating the former competitor model into a more reasonable version that induces recalibration only for some measured SOAs and by including another (asynchrony-contingent) model that is capable of predicting the nonlinearity and asymmetry of recalibration (Fig. 3) while heuristically approximating the causal inference computations. The causal-inference model outperformed the asynchrony-contingent model, as indicated by model evidence (Fig. 4A). Furthermore, model predictions show that the causal-inference model more accurately captures recalibration at large SOAs at both the group (Fig. 4B) and the individual level (Fig. S4).

(2) It does not appear that some key comparisons have been subjected to appropriate inferential statistical tests. Specifically, lines 196-207 - presumably this is the mean (and SD or SE) change in AIC between models across the group of 9 observers. So are these differences actually significant, for example via t-test?

We statistically compared the models using Bayes factors (Fig. 4A). The model evidence for each model was approximated using Variational Bayesian Monte Carlo. Bayes factors provided strong evidence in support of the causal-inference model relative to the other models.

(3) The manuscript tends to gloss over the population-code account of temporal recalibration, which can already provide a quantitative account of how the magnitude of recalibration varies with adapter SOA. This could be better acknowledged, and the features a population code may struggle with (asymmetry?) are considered.

We simulated a population-code model to examine its prediction of the recalibration effect for different adapter SOAs (lines 380–388, Supplement Section 8). The population-code model can predict the nonlinearity of recalibration, i.e., a decreasing recalibration effect as the adapter SOA increases. However, to capture the asymmetry of recalibration effects across auditory-leading and visual-leading adapter stimuli, we would need to assume that the auditory-leading and visual-leading SOAs are represented by neural populations with unequal tuning curves.

(4) The engagement with relevant past literature seems a little thin. Firstly, papers that have applied causal inference modeling to judgments of relative timing are overlooked (see references below). There should be greater clarity regarding how the modelling here builds on or differs from these previous papers (most obviously in terms of additionally modelling the recalibration process, but other details may vary too). Secondly, there is no discussion of previous findings like that in Fujisaki et al.’s seminal work on recalibration, where the spatial overlap of the audio and visual events didn’t seem to matter (although admittedly this was an N = 2 control experiment). This kind of finding would seem relevant to a causal inference account.References:Magnotti JF, Ma WJ and Beauchamp MS (2013) Causal inference of asynchronous audiovisual speech. Front. Psychol. 4:798. doi: 10.3389/fpsyg.2013.00798Sato, Y. (2021). Comparing Bayesian models for simultaneity judgement with different causal assumptions. J. Math. Psychol., 102, 102521.

We have revised the Introduction and Discussion to better situate our study within the existing literature. Specifically, we have incorporated the suggested references (lines 66–69) and provided clearer distinctions on how our modeling approach builds on or differs from previous work on causal-inference models, particularly in terms of modeling the recalibration process (lines 75–79). Additionally, we have discussed findings that might contradict the assumptions of the causal-inference model (lines 405–424).

(5) As a minor point, the model relies on simulation, which may limit its take-up/application by others in the field.

Upon acceptance, we will publicly share the code for all models (simulation and parameter fitting) to enable researchers to adapt and apply these models to their own data.

(6) There is little in the way of reassurance regarding the model’s identifiability and recoverability. The authors might for example consider some parameter recovery simulations or similar.

We conducted a model recovery for each of the six models described in the main text and confirmed that the asynchrony-contingent and causal-inference models are identifiable (Supplement Section 11). Simulations of the asynchrony-correction model were sometimes best fit by causal-inference models, because the latter behaves similarly when the prior of a common cause is set to one.

We also conducted a parameter recovery for the winning model, the causal-inference model with modality-specific precision (Supplement Section 13).

Key parameters, including audiovisual bias , amount of auditory latency noise , amount of visual latency noise , criterion, lapse rate showed satisfactory recovery performance. The less accurate recovery of is likely due to a tradeoff with learning rate .

(7) I don't recall any statements about open science and the availability of code and data.

Upon acceptance of the manuscript, all code (simulation and parameter fitting) and data will be made available on OSF and publicly available.

**Recommendations for the authors:**

**Reviewing Editor (Recommendations For The Authors):**
In addition to the comments below, we would like to offer the following summary based on the discussion between reviewers:The major shortcoming of the work is that there should ideally be a bit more evidence to support the model, over and above a demonstration that it captures important trends and beats an account that was already known to be wrong. We suggest you:(1) Revise the figure legends (Figure 5 and Figure 6E).

We revised all figures and figure legends.

(2) Additionally report model differences in terms of BIC (which will favour the preferred model less under the current analysis);

We now base the model comparison on Bayesian model selection, which approximates and compares model evidence. This method incorporates a strong penalty for model complexity through marginalization over the parameter space (MacKay, 2003).

(3) Move to instead fitting the models multiple times in order to get leave-one-out estimates of best-fitting loglikelihood for each left-out data point (and then sum those for the comparison metric).

Unfortunately, our design is not suitable for cross-validation methods because the model-fitting process is computationally intensive and time-consuming. Each fit of the causal-inference model takes approximately 30 hours, and multiple fits with different initial starting points are required to rule out local minima.

(4) Offering a comparison with a more convincing model for example an atheoretical fit with free parameters for all adapters, e.g., as suggested by Reviewer 3.

We updated the previous competitor model and included an asynchrony-contingent model, which is capable of predicting the nonlinearity of recalibration (Fig. 3). The causal-inference model still outperformed the asynchrony-contingent model (Fig. 4A). Furthermore, model predictions show that only the causal-inference model captures non-zero recalibration effects for long adapter SOAs at both the group level (Fig. 4B) and individual level (Figure S4).

**Reviewer #1 (Recommendations For The Authors):**
A larger sample size would be better.

This study used a within-subject design, which included 9 sessions, totaling 13.5 hours per participant. This extensive data collection allows us to better constrain the model for each participant. Our conclusions are based on the different models’ ability to fit individual data rather than on group statistics.

It would be good to better put the study in the context of spatial ventriloquism, where similar model comparisons have been done over the last ten years and there is a large body of work to connect to.

We now discuss our model in relation to models of cross-modal spatial recalibration in the Introduction (lines 70–78) and Discussion (lines 324–330).

**Reviewer #2 (Recommendations For The Authors):**
Previous authors (e.g., Yarrow et al.,) have described latency shift and criterion change models as providing a good fit of experimental data. Did the authors attempt a criterion shift model in addition to a shift model?

We have considered criterion-shift variants of our atheoretical recalibration models in Supplement Section 1. To summarize the results, we varied two model assumptions: (1) the use of either a Gaussian or an exponential measurement distribution, and (2) recalibration being implemented either as a shift of bias or a criterion. We fit each model variant separately to the ternary TOJ responses of all sessions. Bayesian model comparisons indicated that the bias-shift model with exponential measurement distributions best captured the data of most participants.

Figure 4B - I'm not convinced that the modality-independent uncertainty is anything but a straw man. Models not allowed to be asymmetric do not show asymmetry? (the asymmetry index is irrelevant in the fixed update model as I understand it so it is not surprising the model is identical?).

We included the assumption that temporal uncertainty might be modality-independent for several reasons. First, there is evidence suggesting that a central mechanism governs the precision of temporal-order judgments (Hirsh & Sherrick, 1961), indicating that precision is primarily limited by a central mechanism rather than the sensory channels themselves. Second, from a modeling perspective, it was necessary to test whether an audio-visual temporal bias alone, i.e., assuming modality-independent uncertainty, could introduce asymmetry across adapter SOAs. Additionally, most previous studies implicitly assumed symmetric likelihoods, i.e., modality-independent latency noise, by fitting cumulative Gaussians to the psychometric curves derived from 2AFC-TOJ tasks (Di Luca et al., 2009; Fujisaki et al., 2004; Harrar & Harris, 2005; Keetels & Vroomen, 2007; Navarra et al., 2005; Tanaka et al., 2011; Vatakis et al., 2007, 2008; Vroomen et al., 2004).

Why does a zero SOA adapter shift the pss towards auditory leading? Is this a consequence of the previous day’s conditioning - it’s not clear from the methods whether all listeners had the same SOA conditioning sequence across days.

The auditory-leading recalibration effect for an adapter SOA of zero has been consistently reported in previous studies (e.g., Fujisaki et al., 2004; Vroomen et al., 2004). This effect symbolizes the asymmetry in recalibration. This asymmetry can be explained by differences across modalities in the noisiness of the latencies (Figure 5C) in combination with audiovisual temporal bias (Figure S8).

We added details about the order of testing to the Methods section (lines 456–457).

**Reviewer #3 (Recommendations For The Authors):**
Abstract“Our results indicate that human observers employ causal-inference-based percepts to recalibrate cross-modal temporal perception” Your results indicate this is plausible. However, this statement (basically repeated at the end of the intro and again in the discussion) is - in my opinion - too strong.

We have revised the statement as suggested.

Intro and laterWithin the wider literature on relative timing perception, the temporal order judgement (TOJ) task refers to a task with just two response options. Tasks with three response options, as employed here, are typically referred to as ternary judgments. I would suggest language consistent with the existing literature (or if not, the contrast to standard usage could be clarified).Ref: Ulrich, R. (1987). Threshold models of temporal-order judgments evaluated by a ternary response task. Percept. Psychophys., 42, 224-239.

We revised the term for the task as suggested throughout the manuscript.

Results, 2.2.2“However, temporal precision might not be due to the variability of arrival latency.” Indeed, although there is some recent evidence that it might be.Ref: Yarrow, K., Kohl, C, Segasby, T., Kaur Bansal, R., Rowe, P., & Arnold, D.H. Neural-latency noise places limits on human sensitivity to the timing of events. Cognition, 222, 105012 (2022).

We included the reference as suggested (lines 245–248).

Methods, 4.3.Should there be some information here about the order of adaptation sessions (e.g. random for each observer)?

We added details about the order of testing to the Methods section (lines 456–457).

Supplemental material section 1.Here, you test whether the changes resulting from recalibration look more like a shift of the entire psychometric function or an expansion of the psychometric function on one side (most straightforwardly compatible with a change of one decision criterion). Fine, but the way you have done this is odd, because you have introduced a further difference in the models (Gaussian vs. exponential latency noise) so that you cannot actually conclude that the trend towards a win for the bias-shift model is simply down to the bias vs. criterion difference. It could just as easily be down to the different shapes of psychometric functions that the two models can predict (with the exponential noise model permitting asymmetry in slopes). There seems to be no reason that this comparison cannot be made entirely within the exponential noise framework (by a very simple reparameterization that focuses on the two boundaries rather than the midpoint and extent of the decision window). Then, you would be focusing entirely on the question of interest. It would also equate model parameters, removing any reliance on asymptotic assumptions being met for AIC.

We revised our exploration of atheoretical recalibration models. To summarize the results, we varied two model assumptions: (1) the use of either a Gaussian or an exponential measurement distribution, and (2) recalibration being implemented either as a shift of the cross-modal temporal bias or as a shift of the criterion. We fit each model separately to the ternary TOJ responses of all sessions. Bayesian model comparisons indicated that the bias-shift model with exponential measurement distributions best described the data of most participants.

References

Di Luca, M., Machulla, T.-K., & Ernst, M. O. (2009). Recalibration of multisensory simultaneity: cross-modal transfer coincides with a change in perceptual latency. Journal of Vision, 9(12), Article 7.

Fujisaki, W., Shimojo, S., Kashino, M., & Nishida, S. ’ya. (2004). Recalibration of audiovisual simultaneity. Nature Neuroscience, 7(7), 773–778.

Harrar, V., & Harris, L. R. (2005). Simultaneity constancy: detecting events with touch and vision. Experimental Brain Research. Experimentelle Hirnforschung. Experimentation Cerebrale, 166(3-4), 465–473.

Hirsh, I. J., & Sherrick, C. E., Jr. (1961). Perceived order in different sense modalities. Journal of Experimental Psychology, 62(5), 423–432.

Keetels, M., & Vroomen, J. (2007). No effect of auditory-visual spatial disparity on temporal recalibration. Experimental Brain Research. Experimentelle Hirnforschung. Experimentation Cerebrale, 182(4), 559–565.

MacKay, D. J. (2003). Information theory, inference and learning algorithms.

Navarra, J., Vatakis, A., Zampini, M., Soto-Faraco, S., Humphreys, W., & Spence, C. (2005). Exposure to asynchronous audiovisual speech extends the temporal window for audiovisual integration. Brain Research. Cognitive Brain Research, 25(2), 499–507.

Tanaka, A., Asakawa, K., & Imai, H. (2011). The change in perceptual synchrony between auditory and visual speech after exposure to asynchronous speech. Neuroreport, 22(14), 684–688.

Vatakis, A., Navarra, J., Soto-Faraco, S., & Spence, C. (2007). Temporal recalibration during asynchronous audiovisual speech perception. Experimental Brain Research. Experimentelle Hirnforschung. Experimentation Cerebrale, 181(1), 173–181.

Vatakis, A., Navarra, J., Soto-Faraco, S., & Spence, C. (2008). Audiovisual temporal adaptation of speech: temporal order versus simultaneity judgments. Experimental Brain Research. Experimentelle Hirnforschung. Experimentation Cerebrale, 185(3), 521–529.

Vroomen, J., Keetels, M., de Gelder, B., & Bertelson, P. (2004). Recalibration of temporal order perception by exposure to audio-visual asynchrony. Brain Research. Cognitive Brain Research, 22(1), 32–35.